# SARAD: Spatial Association-Aware Anomaly Detection and Diagnosis for Multivariate Time Series

**Zhihao Dai**
Department of Computer Science
University of Warwick
Coventry, UK
zhihao.dai@warwick.ac.uk

**Ligang He**[*]
Department of Computer Science
University of Warwick
Coventry, UK
ligang.he@warwick.ac.uk

**Shuang-Hua Yang**
Department of Computer Science
University of Reading
Reading, UK
shuang-hua.yang@reading.ac.uk

**Matthew Leeke**
School of Computer Science
University of Birmingham
Birmingham, UK
m.leeke@bham.ac.uk

## Abstract

Anomaly detection in time series data is fundamental to the design, deployment, and evaluation of industrial control systems. Temporal modeling has been the natural focus of anomaly detection approaches for time series data. However, the focus on temporal modeling can obscure or dilute the spatial information that can be used to capture complex interactions in multivariate time series. In this paper, we propose SARAD, an approach that leverages spatial information beyond data autoencoding errors to improve the detection and diagnosis of anomalies. SARAD trains a Transformer to learn the spatial associations, the pairwise inter-feature relationships which ubiquitously characterize such feedback-controlled systems. As new associations form and old ones dissolve, SARAD applies subseries division to capture their changes over time. Anomalies exhibit association descending patterns, a key phenomenon we exclusively observe and attribute to the disruptive nature of anomalies detaching anomalous features from others. To exploit the phenomenon and yet dismiss non-anomalous descent, SARAD performs anomaly detection via autoencoding in the association space. We present experimental results to demonstrate that SARAD achieves state-of-the-art performance, providing robust anomaly detection and a nuanced understanding of anomalous events.

## 1 Introduction

Time series anomaly detection is critical for industrial automation (Rieth et al., 2018), intrusion detection (Mathur and Tippenhauer, 2016), and healthcare sensing (Goldberger et al., 2000). Anomaly detection in these contexts is typically treated as an unsupervised learning problem, owing to the novelty of anomalies and the scarcity of labeled anomalies.

Temporal modeling is the mainstream basis of current time series anomaly detectors. By learning the dependencies between discrete time steps, temporal modeling can pinpoint the time spans of anomalies. During anomalies, unseen and peculiar temporal dependence patterns degrade data autoencoding (Wang et al., 2023b) or autoregression (Zhao et al., 2020) performance, thereby enabling detection. Alternatively, irregular temporal representations can be driven to breach learned enclosing

---

[*]Corresponding Author

38th Conference on Neural Information Processing Systems (NeurIPS 2024).

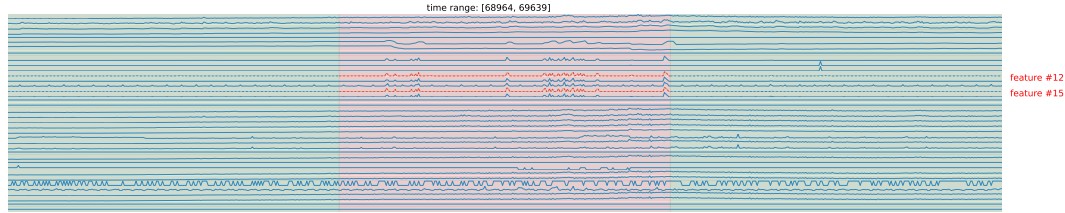

(a) Raw time series before, during, and after an anomaly $p_i$.

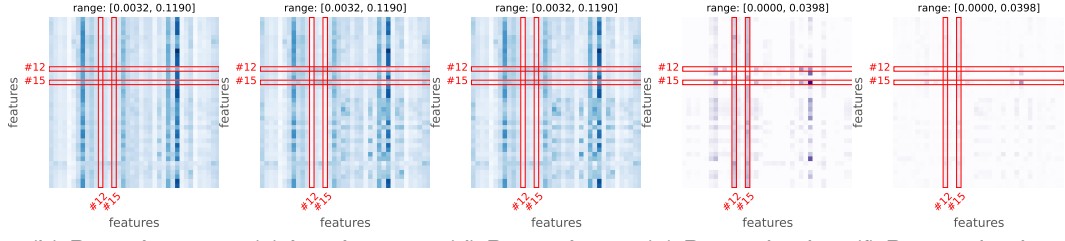

(b) Pre-$p_i$'s $A$. (c) In-$p_i$'s $A$. (d) Post-$p_i$'s $A$. (e) Pre-reduction. (f) Post-reduction.

Figure 1: Spatial associations captured by Transformer on a service monitoring benchmark. 1a shows the raw time series right before, during, and after an anomaly (colored in red). Association mapping $A_h^L$ by final $L$-th layer's MHSA are averaged across heads to derive $A$ before (1b), during (1c), and after (1d) the anomaly. Darker cells have larger values. Anomalous features (#12 and #15) are highlighted with red bounding boxes. The reduction-only changes from before or after the anomaly to during the anomaly are shown in 1e and 1f, i.e., ReLU($A_{\text{pre}} - A_{\text{in}}$) and ReLU($A_{\text{post}} - A_{\text{in}}$). The anomaly leads to association reductions on anomalous features, prominently column-wise on $A$.

hyperspheres (Shen et al., 2020), resulting in high anomaly scores that enable detection. Despite its temporal precision in anomaly detection, temporal modeling either assumes feature independence or combines variables of diverse physical nature, the former simplifying the modeling and the latter mitigating the multicollinearity issue. Such assumptions lead to either omission or dilution of spatial information crucial to anomaly detection. Specifically, it overlooks the long-time-range spatial associations, the relationships between various features which ubiquitously characterize normal behaviors of multivariate time series. Where anomaly detection pinpoints the temporal locations of an anomaly, anomaly diagnosis identifies the spatial locations, i.e., the anomalous feature set, of an anomaly. Temporal methods also restrict diagnostic capabilities, as the lack of spatial information mismatches autoencoding-based or autoregression-based anomaly criterion, which de facto measures temporal novelty, with its objective of capturing spatial novelty.

Furthermore, time series anomalies frequently dissolve spatial associations, motivating anomaly detection in the association space. Using an vanilla Transformer (Vaswani et al., 2017), we investigate the changes in spatial associations throughout anomalies. Applied on transposed time windows (the spatial dimension comes before the temporal), an encoder-only Transformer is trained to minimize reconstruction errors on unlabeled $N$-variate time series and, by doing so, learns to model the multivariate series spatially via the Multi-Head Self-Attention (MHSA) illustrated in Figure 2. MHSA at each stacked $l$-th layer computes an intermediate association mapping $A_h^l \in \mathbb{R}^{N \times N}$ per $h$-th head, mapping back input $\mathcal{X}$ to produce attention scores. The last layer's mapping $A_h^L$ thus effectively captures the contributions of $k$-th feature to the reconstruction of $j$-th feature at each location $(j, k)$, not least for its architectural proximity to the reconstructed output. Recent research (Liu et al., 2024) also highlights the important role MHSA plays in capturing the inter-feature associative relationships when applied on the multi-variate dimension. As new associations emerge and old ones dissolve over time, Figure 1 shows the association changes on a real-world benchmark. We observe that anomalies exhibit reductions for anomalous features, a phenomenon we herein coin as **Spatial**

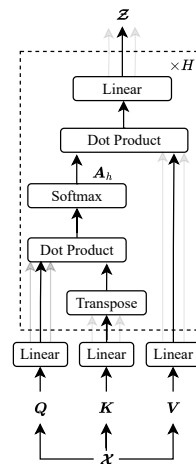

Figure 2: MHSA.

**Association Reduction (SAR)**. The rationale is that anomalies either originate from or result in dissolution of pre-existing associations, detaching anomalous features from their non-anomalous counterparts. Additionally, we make the observation that SAR is most prominent column-wise on $A_h^L$, since each $j$-th column characterizes the dropouts of $j$ from associating with other, mostly non-anomalous, features. Due to lack of explicit spatial information, temporal modeling is inadequate for exploiting SAR. More examples are given in Appendix C.

From a spatial modeling perspective, we propose SARAD to leverage spatial information and to exploit SAR for enabling robust time series anomaly detection and diagnosis. For quantifying anomalous spatial novelty in the data space, we train a Transformer on transposed time windows as an autoencoder. To capture the spatial association progression, the reduction-only changes of associations over time, the data reconstruction divides the input window by time into two halves to be processed in parallel. Consequently, the progression is the non-negative backward difference of the intermediate association mappings via MHSA. Subseries division circumvents memory storage of latest association mappings and enables time window shuffling during training, which reduces order bias, enhances generalization, and prevents catastrophic forgetting. For quantifying anomalous reduction novelty, we train a Multi-Layer Perceptron (MLP) as an antuencoder on progression in the association space. Whereas progression encompass all association reduction, autoencoding rules out those not caused by anomalies. The reconstruction errors via the data module measure data-only anomalous deviation from expected system behaviors and falter when such deviations are not prominent, e.g., at the start of an anomaly. The reconstruction errors via the progression module are sensitive to change in spatial associations, thus complementing the former. We develop a joint anomaly detection criterion that combines both. Experiments show SARAD delivers state-of-the-art detection and diagnosis performance with architectural elegance. Code is available at `https://github.com/daidahao/SARAD/`. We summarize our contributions as follows.

- We reveal and extract spatial association descending patterns of time series anomalies with a bespoke Transformer and subseries division. The former learns the pairwise inter-feature associations via autoencoding in the data space and the latter enables shuffled autoencoding training and memory-less progression aggregation.

- We propose progression autoencoding to quantify anomalous descent in the association space and a joint detection criterion in both data and association spaces, which complement each other.

- Experimentally, SARAD performs state-of-the-art anomaly detection and diagnosis on multivariate time series and ablation studies support our design choices.

## 2 Related Work

Influenced by the dominance of temporal modeling in time series forecasting (Wang et al., 2023a; Zhang et al., 2023; Wu et al., 2021), temporal modeling is also prevalent in time series anomaly detection. Recurrent neural networks such as LSTM (Hochreiter and Schmidhuber, 1997) have innate capabilities for handling sequential data. These approaches use hidden states for past input memorization, enabling detection (Li et al., 2019; Malhotra et al., 2015) and diagnosis (Qian et al., 2021). Transformer (Vaswani et al., 2017) network is widely adopted (Fan et al., 2023; Xu et al., 2022) approach that is commonly applied to model temporal associations between different time points using its attention mechanism. Linear regression (Zeng et al., 2023) and MLP (Wang et al., 2024; Audibert et al., 2020) directly model temporal dependencies. TranAD (Tuli et al., 2022) replaces the MLP in Audibert et al. (2020) with a Transformer, making the detection criterion more robust through its adversarial training paradigm. Temporal modeling, however, is restricted by the exceptionally small receptive field in time and adversely impacted by the timestamp misalignment across features. In the context of anomaly detection, temporal modeling helps capture anomalous temporal associations (Xu et al., 2022; Yang et al., 2023), but offers limited detection capabilities in absence of spatial information. In a diagnostic context, temporal detectors mismatch anomaly criterion of temporal novelty with spatial interpretation.

Spatial associations characterize the multivariate time series commonly found in such supervisory systems for industrial control. The relationships range from strongly correlated, e.g., due to spatial proximity, to fully independent, e.g., due to mechanical disconnection. For forecasting, iTrans-former (Liu et al., 2024) applies Transformer on the transposed time series to enable direct spatial

modeling. Crossformer (Zhang and Yan, 2023) screens the time series through custom Two-Stage Attention layers for more efficient spatial modeling. In terms of detection, GDN (Deng and Hooi, 2021) learns a directed graph of features for the prediction of last time points, whose errors serve as anomaly scores. GDN is partially limited by a mismatch between its singe-timestamp prediction target and the prevalent range-wise anomalies as well as unstable Top-K node selection during training. InterFusion (Li et al., 2021) learns compressed spatial and temporal dependencies, using a hierarchical Variational Auto-Encoder (Kingma and Welling, 2014) to reconstruct the series. Neither inspects temporal changes in associations throughout anomalies.

On another front, Isolation Forest (IF) models build a binary decision tree ensemble by partitioning either the data space (Liu et al., 2008) or the deep embedding space (Xu et al., 2023) formed by randomized neural networks. They are constrained by the lack of temporal and spatial (in the former case) or spatial (in the latter case) information, and their anomaly scores are not reflective of the degrees of anomalies.

We emphasize anomalous association descending patterns towards better time series detection and diagnosis. Different from previous work, we explicitly utilize the reduction in spatial associations over time during an anomaly, an insight we derived from the cyber-physical defense space. Dynamic watermarking (Satchidanandan and Kumar, 2017) and similar defense techniques (Dai et al., 2023) overlay actuation with private signals to reveal attacks resulting in correlational breakdowns. While their approaches are intrusive and actively alter system behaviors, our detector remains non-intrusive, passively monitors the spatial associations, and is applicable to any supervisory system.

We refer to spatiality in this work as the multi-dimensional vector nature inherent to multivariate time series data. The terminology is also used in literature on time series related tasks (Gangopadhyay et al., 2021; Zheng et al., 2023). We note that spatiality may carry different meanings in other AI contexts, such as geographic positions or characteristics on Earth. We differentiate those meanings from our definition of spatiality, which traces its root to the spatial distribution of sensors and actuators in control systems where time series are routinely collected.

## 3 Method

The problems of anomaly detection and diagnosis are specified as follows.

**Anomaly Detection** Given a $N$-feature time series $\mathcal{T} = \{x^1, \cdots, x^N\}$ where $x^n \in \mathbb{R}^T$ is of the same length $T$, the objective is to predict the anomaly label $y_t \in \{0, 1\}$ at each timestamp $t$.

**Anomaly Diagnosis** Given the same time series $\mathcal{T}$, the objective is to predict the diagnosis label $g_t \subseteq [N]$, the set of anomalous features at each timestamp $t$.

### 3.1 Overview

SARAD comprises two sequential modules; a Transformer for time series data reconstruction and a MLP network for spatial progression reconstruction. Table 1 decomposes the system framework of SARAD. The Transformer temporally divides by 2 and reconstructs the input time series to learn pairwise inter-feature associations and to enable order-free memory-efficient progression aggregation. The MLP reconstructs the aggregated progression to quantify anomalous association reduction while dismissing non-anomalous reduction. Towards robust anomaly detection, reconstruction errors from the two modules jointly serve as a criterion, sensitive to data-only anomalous deviation and anomalous association reduction.

### 3.2 Data Reconstruction

In light of restricted capabilities of temporal detectors, here we adapt Transformer to spatially reconstruct the series data. The data module contains two components, Subseries Split & Merge and Subseries Reconstruction, shown in the first and second columns in Table 1. The former wraps around the second by temporally splitting a multivariate input series in half at its beginning and temporally merging at its end. Subseries division enables capturing of spatial progression within a single time window. Without the former, the model must store in memory the last association mappings at each step and keep to the time ordering during training, which is prone to overfitting and

Table 1: SARAD is a composition of two modules and three components: data reconstruction (subseries split and merge, and subseries reconstruction) and spatial progression reconstruction.

| | Data Reconstruction | | Progression Recon. |
|---|---|---|---|
| | Subseries Split & Merge | Subseries Recon. | |
| Architecture | - | Encoder-only Transformer | MLP |
| Diagram | 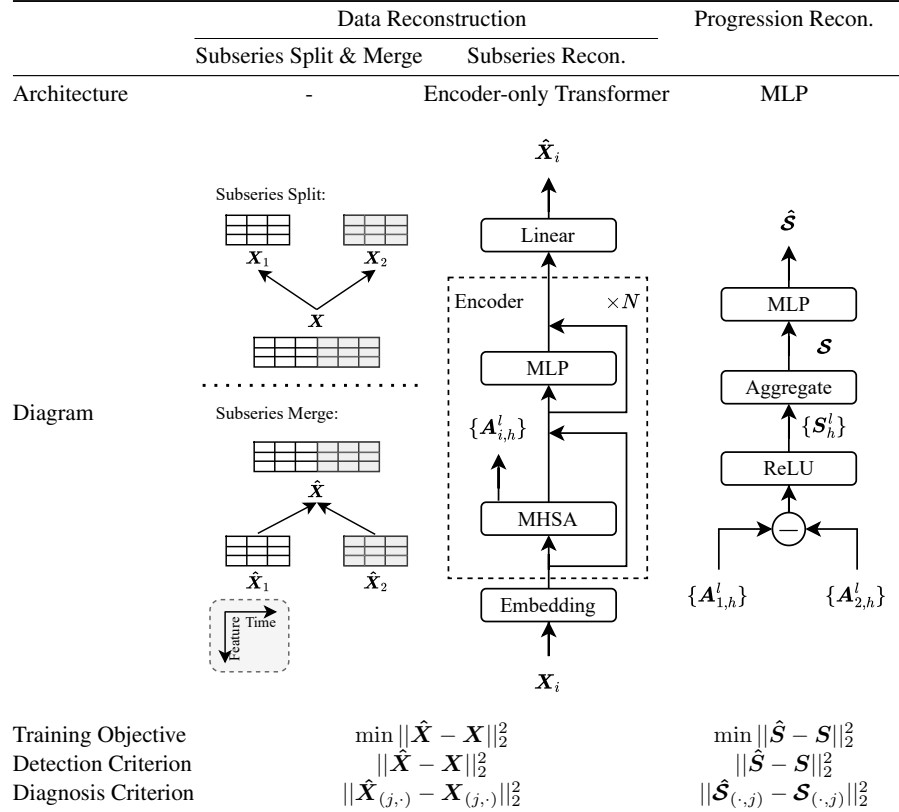 | | |
| Training Objective | $\min \|\hat{\boldsymbol{X}} - \boldsymbol{X}\|_2^2$ | | $\min \|\hat{\boldsymbol{S}} - \boldsymbol{S}\|_2^2$ |
| Detection Criterion | $\|\hat{\boldsymbol{X}} - \boldsymbol{X}\|_2^2$ | | $\|\hat{\boldsymbol{S}} - \boldsymbol{S}\|_2^2$ |
| Diagnosis Criterion | $\|\hat{\boldsymbol{X}}_{(j,\cdot)} - \boldsymbol{X}_{(j,\cdot)}\|_2^2$ | | $\|\hat{\boldsymbol{\mathcal{S}}}_{(\cdot,j)} - \boldsymbol{\mathcal{S}}_{(\cdot,j)}\|_2^2$ |

catastrophic forgetting. The latter is an encoder-only Transformer composed of an embedding layer, a $L$-layer attention-based encoder, and finally a linear projection layer. Encoder-only Transformers are commonly found in Transformer-based detectors (Kang and Kang, 2024; Kim et al., 2023) due to its simplicity and the length uniformity of the target output, i.e., the reconstructed series. Ours exclusively models spatial associations, unconventional to the aforenamed detectors and most temporal forecasters (Nie et al., 2023; Zhou et al., 2022; Liu et al., 2022) and yet more aligned with recent spatial-aware forecasters (Liu et al., 2024; Zhang and Yan, 2023). At each encoding layer, MHSA computes pairwise association mappings, a representation of inter-feature dependencies which ubiquitously characterize the multivariate time series and are crucial to anomaly detection.

**Subseries Split and Merge**  We suppose the input series is a time window $\boldsymbol{X} \in \mathbb{R}^{2W \times N}$ of length $2W$, where $2$ is for the convenience of a temporal split. Before reconstruction, $\boldsymbol{X}$ is split into two half multivariate subseries of equal temporal length: $\boldsymbol{X} = \{\boldsymbol{X}_1 \in \mathbb{R}^{W \times N}, \boldsymbol{X}_2 \in \mathbb{R}^{W \times N}\}$. After subseries reconstruction, the two reconstructed subseries are concatenated to form the full reconstructed $\hat{\boldsymbol{X}} = \{\hat{\boldsymbol{X}}_1 \in \mathbb{R}^{W \times N}, \hat{\boldsymbol{X}}_2 \in \mathbb{R}^{W \times N}\}$.

**Embedding**  To lead subseries reconstruction, each $\boldsymbol{X}_i$ is embedded as $\boldsymbol{\mathcal{X}}_i^0 = \boldsymbol{E}_i + \boldsymbol{M}$, wherein $\boldsymbol{E}_i = \text{Linear}(\boldsymbol{X}_i^T) \in \mathbb{R}^{N \times D}$ and $\boldsymbol{M} = \{\boldsymbol{m}_i \in \mathbb{R}^D | i \in [N]\}$ is a learnable feature-level embedding.

**Spatial-Aware Encoding**  A stack of $L$ Transformer encoding layers is used to encode the series in the $D$-length attention space. Each layer is stacked with MHSA and MLP with residual connections:

$$\boldsymbol{\mathcal{Z}}_i^l = \text{LN}(\text{MHSA}(\boldsymbol{\mathcal{X}}_i^{l-1}) + \boldsymbol{\mathcal{X}}_i^{l-1}), \ \boldsymbol{\mathcal{X}}_i^l = \text{LN}(\text{MLP}(\boldsymbol{\mathcal{Z}}_i^l) + \boldsymbol{\mathcal{Z}}_i^l) \quad (1)$$

where $\boldsymbol{\mathcal{X}}_i^{l-1}, \boldsymbol{\mathcal{Z}}_i^l, \boldsymbol{\mathcal{X}}_i^l \in \mathbb{R}^{N \times D}$ are $l-1$ layer's output, $l$-th layer's hidden state and output respectively and $LN(\cdot)$ is Layer Normalization (Ba et al., 2016). It is spatial-aware because MHSA explicitly learns a pairwise inter-feature association mapping to exchange information in between feature

representations. Notably within the MHSA with $H$ heads as shown in Figure 2, each $(j, k)$-th element on the association mapping $\boldsymbol{A}_h \in \mathbb{R}^{N \times N}$ of its $h$-th head computes how much $j$-th feature's attention scores should originate from the $k$-th feature's key. From a broader perspective of data reconstruction, it is the quantification of the residual impact of $k$-th feature's originals on the $j$-th feature's reconstructions. We refactor MHSA implementation to enable parallel encoding of subseries.

**Linear Projection** Output from the last encoding layer $\boldsymbol{\mathcal{X}}_i^L$ is linearly projected and transposed to derive the reconstructed subseries $\hat{\boldsymbol{X}}_i \in \mathbb{R}^{W \times N}$.

## 3.3 Spatial Progression Reconstruction

To exploit SAR caused by anomalies, the module first extracts and aggregates the spatial progression, the non-negative backward difference of the association mappings via MHSA. In line with general anomaly detectors (Aggarwal, 2013), the module conducts autoencoding in the association space to quantify anomalous SAR and to dismiss non-anomalous SAR. Anomalous SAR occurs when, say, a compromised sensor's readings are no longer correlating with its spatially adjacent or mechanically related counterparts.

**Association Progression** We define association progression $\boldsymbol{S}_h^l \in \mathbb{R}^{N \times N}$ at the $h$-th attention head in the $l$-th layer to be the non-negative backward difference in association mappings $\{\boldsymbol{A}_{1,h}^l, \boldsymbol{A}_{2,h}^l\}$:

$$\boldsymbol{S}_h^l = \text{ReLU}(\boldsymbol{A}_{1,h}^l - \boldsymbol{A}_{2,h}^l) \tag{2}$$

where $\text{ReLU}(\cdot)$ passes through only non-negative values and outputs zeros otherwise.

**Progression Aggregation** To center the detection on association dropouts, we aggregate the column sums of progression $\boldsymbol{S}_h^l$ from all attention heads in the final $L$-th layer to form $\boldsymbol{\mathcal{S}} \in \mathbb{R}^{H \times N}$:

$$\boldsymbol{\mathcal{S}} = \{\sum_{j=1}^{N} \boldsymbol{S}_{h,(j,k)}^L | h \in [H], k \in [N]\} \tag{3}$$

We recall from the data module, each $k$-th column in $\boldsymbol{A}_{i,h}^l$ quantifies the impact of $k$-th feature on all features' reconstruction. Taking the sum per each $k$-th column, we measure with $\boldsymbol{\mathcal{S}}$ the dropout rates of $k$ from participating all features' reconstruction. As we have observed in Section 1, SAR at the column level is indicative of time series anomalies, more so than at the row level. The last layer's progression is focused not least for its proximity to the final reconstructed output, whereafter no more information is exchanged between features. Liu et al. (2024) manifests that final layer's mappings resemble closely with the inter-feature correlations of the target, in our case, the reconstructed.

**Autoencoding** With $\boldsymbol{\mathcal{S}}$ flattened as an one-dimensional vector, a 2-layer MLP is trained to output the reconstructed and reshaped $\hat{\boldsymbol{\mathcal{S}}} \in \mathbb{R}^{H \times N}$, synchronously with the data module training.

## 3.4 Joint Training and Anomaly Detection

**Training Objective** We train an end-to-end model with a joint minimization objective:

$$L_R = ||\hat{\boldsymbol{X}} - \boldsymbol{X}||_2^2, \; L_S = ||\hat{\boldsymbol{\mathcal{S}}} - \boldsymbol{\mathcal{S}}||_2^2, \; L = L_R + \lambda_{L_S} L_S \tag{4}$$

where $L_R$ is the data reconstruction loss, $L_S$ the progression reconstruction loss, and $\lambda_{L_S}$ a weight hyper-parameter. Gradients are stopped from flowing into $\boldsymbol{\mathcal{S}}$ to prevent updates to the data module and collapses in association representation. We are training two anomaly detectors simultaneously, one working in the original data space, the other in the spatial progression space.

**Anomaly Detection Criterion** For an input series $\boldsymbol{X}$, the anomaly score $s$ is a scalar defined to be:

$$r = ||\hat{\boldsymbol{X}} - \boldsymbol{X}||_2^2, \; p = ||\hat{\boldsymbol{\mathcal{S}}} - \boldsymbol{\mathcal{S}}||_2^2, \; s = (r - \mu_r)/\sigma_r + (p - \mu_p)/\sigma_p \tag{5}$$

where $r$ is the data reconstruction error, $p$ the progression reconstruction error, $\mu_r, \mu_p$ the means of $r, p$ on the validation set, and $\sigma_r, \sigma_p$ the standard deviation of $r, p$. The criterion takes into account the normalized errors in the data space and the progression space, each of which quantifies the anomalous magnitude in respective spaces and complment the other.

Table 2: Statistics of the main datasets.

| Dataset | Features | Training Set | Test Set | Anomalies | | Lengths | | | Sampling Period |
| | | | | Count | Ratio | min | med | max | |
|---|---|---|---|---|---|---|---|---|---|
| SMD | 38 | 708,405 | 708,420 | 327 | 4.16% | 2 | 11 | 3,161 | 1 min |
| PSM | 25 | 132,481 | 87,841 | 71 | 27.73% | 1 | 5 | 8,861 | 1 min |
| SWaT | 51 | 496,800 | 449,919 | 34 | 12.02% | 101 | 447 | 35,900 | 1 sec |
| HAI | 79 | 921,603 | 402,005 | 50 | 2.23% | 17 | 162.5 | 422 | 1 sec |

**Anomaly Diagnosis Criterion**   For an input series $\boldsymbol{X}$ and its $j$-th feature, its anomaly score $s_j$ is a scalar defined to be:

$$s_j = r_j = ||\hat{\boldsymbol{X}}_{(j,\cdot)} - \boldsymbol{X}_{(j,\cdot)}||_2^2, \tag{6}$$

where $r_j$ is feature $j$'s data reconstruction error. The criterion is sensitive to spatial novelty.

# 4   Experiments

SARAD is compared against state-of-the-art detectors on real-world benchmarks for detection and diagnosis, the latter only when diagnostic labels are available.

## 4.1   Experimental Setup

**Datasets**   We evaluate on four real-world datasets collected under industrial control and service monitoring settings. These dataset are: 1) Server Machine Dataset (**SMD**) (Su et al., 2019b,a), 2) Pooled Server Metrics (**PSM**) dataset (Abdulaal et al., 2021a,b) 3) Secure Water Treatment (**SWaT**) dataset (Mathur and Tippenhauer, 2016; iTrust, 2023), and 4) Hardware-In-the-Loop-based Augmented ICS (**HAI**) dataset (Shin et al., 2021b,a). All training sets contain only unlabeled data and the test sets contain data with anomaly labels. Anomalies range from service outages to external cyber-physical attacks. We summarize the statistics of the datasets in Table 2. Descriptions of each dataset are detailed in Appendix E.

**Detection Metrics**   Real-world benchmarks are rife with range-wise anomalies spanning consecutive time points (Wagner et al., 2023). We use the range-based metrics proposed in (Paparrizos et al., 2022). Compared against their point-based counterparts, they provide robustness to labeling delay and scoring noises as well as performant detector separability and series consistency. We compute the threshold-independent AUC-ROC and AUC-PR scores to be rid of thresholding impact and fully parameter-free Volume Under the Surface (VUS) AUC-ROC and AUC-PR scores. Full details are discussed in Appendix I.

**Diagnosis Metrics**   Consistent with previous works (Tuli et al., 2022; Zhao et al., 2020), we use common metrics such as Hit Rate (HR) (Su et al., 2019b) and Normalized Discounted Cumulative Gain (NDCG) (Järvelin and Kekäläinen, 2002) where diagnosis labels are available. At the range level, we measure the **I**nterpretation **S**core (IPS) initially proposed in Li et al. (2021) and here expanded to fit the $P\%$ parameterization. Full details are discussed in Appendix J.

**Baselines**   We compare SARAD against state-of-the-art anomaly detection baselines, including Isolation Forest-based IF (Liu et al., 2008), Deep IF (DIF) (Xu et al., 2023); MLP-based USAD (Audibert et al., 2020); graph-based GDN (Deng and Hooi, 2021); LSTM-based MAD-GAN (Li et al., 2019); CNN-based DiffAD (Xiao et al., 2023); and Transformer-based TranAD (Tuli et al., 2022), ATF-UAD (Fan et al., 2023), AT (Xu et al., 2022), DCdetector (Yang et al., 2023). Noticeably, GDN employs explicit spatial modeling in its graph construction although spatial associations are not directly involved in anoamly scoring. MAD-GAN emphasizes on anomaly detection within cyber-physical systems. All baselines are trained using official implementations where available and recommended hyperparameters from respectively papers are used.

Table 3: Anomaly detection performance. Threshold-independent AUC-ROC and AUC-PR metrics and fully parameter-free VUS-ROC and VUS-PR metrics are reported. All values are average percentages from five random seeds. The best values are in **bold** and the second best underlined.

| Method | SMD | | | | PSM | | | | SWaT | | | | HAI | | | |
|---|---|---|---|---|---|---|---|---|---|---|---|---|---|---|---|---|
| | $A_{ROC}$ | $A_{PR}$ | $V_{ROC}$ | $V_{PR}$ | $A_{ROC}$ | $A_{PR}$ | $V_{ROC}$ | $V_{PR}$ | $A_{ROC}$ | $A_{PR}$ | $V_{ROC}$ | $V_{PR}$ | $A_{ROC}$ | $A_{PR}$ | $V_{ROC}$ | $V_{PR}$ |
| IF | 53.81 | 7.27 | 53.56 | 7.25 | 58.08 | **41.51** | 57.99 | **41.48** | 86.11 | 66.52 | 84.39 | 63.57 | 72.90 | 10.03 | 71.65 | 9.81 |
| DIF | 60.27 | 10.30 | 59.84 | 10.23 | 52.00 | 36.61 | 51.88 | 36.55 | **89.38** | **73.19** | **87.88** | 70.54 | 82.10 | 35.86 | 81.13 | 34.16 |
| TranAD | 46.86 | 5.92 | 46.54 | 5.88 | 50.20 | 35.22 | 49.47 | 35.19 | 47.78 | 17.65 | 47.13 | 17.57 | 75.60 | 25.80 | 75.06 | 25.41 |
| ATF-UAD | 43.41 | 4.98 | 43.10 | 4.97 | 46.44 | 33.20 | 46.03 | 33.18 | 55.18 | 20.66 | 54.35 | 20.58 | 70.56 | 22.47 | 69.81 | 22.06 |
| AT | 50.01 | 5.42 | 49.97 | 5.36 | 37.66 | 26.49 | 36.82 | 26.47 | 46.77 | 12.95 | 46.45 | 12.79 | 47.41 | 5.85 | 47.24 | 5.85 |
| DCdetector | 49.47 | 4.51 | 49.10 | 4.50 | 45.94 | 24.76 | 46.01 | 24.82 | 50.80 | 14.47 | 50.76 | 14.37 | N/A | N/A | N/A | N/A |
| USAD | 50.20 | 6.93 | 50.01 | 6.91 | 42.45 | 33.23 | 42.30 | 33.20 | 80.36 | 60.06 | 78.53 | 57.33 | 72.59 | 23.27 | 71.84 | 22.73 |
| GDN | 66.37 | 9.40 | 66.07 | 9.34 | **63.51** | 40.66 | **63.13** | 40.53 | 79.30 | 28.12 | 78.79 | 28.17 | 84.79 | 35.82 | 84.03 | 35.05 |
| MAD-GAN | 64.35 | 9.77 | 64.16 | 9.74 | 57.50 | 40.08 | 57.37 | 40.03 | 86.51 | 61.95 | 86.10 | 62.03 | 84.92 | 49.06 | 84.09 | 48.14 |
| DiffAD | 58.71 | 7.22 | 58.40 | 7.19 | 51.60 | 32.10 | 51.02 | 32.01 | 27.02 | 9.22 | 26.45 | 9.21 | 86.96 | 21.95 | 86.25 | 21.74 |
| **Ours** | **79.97** | **15.09** | **79.67** | **15.02** | 61.87 | 41.06 | 61.77 | 41.01 | 88.29 | 72.90 | 87.52 | **70.68** | **96.87** | **67.78** | **96.17** | **64.70** |

## 4.2 Results

**Anomaly Detection** Table 3 shows the anomaly detection performance in metrics defined in Section 4.1. It demonstrates that, despite its architectural elegance, SARAD either outperforms all baselines by significant margins on the threshold-independent VUS-ROC scores (SMD: $+15.51\%$ MAD-GAN, HAI: $+9.92\%$ DiffAD) or performs on par with current best detectors (PSM: $-1.36\%$ GDN, SWaT: $-0.36\%$ DIF). IF scrutinizes the distributional shifts of anomalies with random data partitions and delivers consistent performance across datasets. DIF extends IF into randomized deep representation spaces and archives decent improvements due to more flexible partitions and temporally local information extraction via dilated convolutions. Temporal modeling methods such as DiffAD, ATF-UAD, and AT rely solely or heavily on reconstruction errors and when the errors do not correspond the the underlying anomalies their performance plummet. Adversarial training in USAD and MAD-GAN amplifies reconstruction errors of anomalies to mitigate but not eliminate such issues and thus suffer less performance drops. In contrast, our SARAD additionally accounts for the SAR frequent with anomalies and independent of data distributional shifts, thus outperforming all. SARAD also overpasses GDN, which despite its explicit spatial modeling adopts prediction errors as its sole detection criterion, limiting its performance. SARAD's top performance on SWaT and HAI underlines its ability to unravel complex spatial associations even in complex large-scale systems. Standard deviations of Table 3 are reported in Appendix K.

**Anomaly Diagnosis** Table 4 shows the anomaly diagnosis performance in metrics defined in Section 4.1. DiffAD uses a subset of SMD features and thus is discarded from comparisons for fairness. SARAD outperforms baselines on the point-based HR@150% (SMD: $+26.67\%$ TranAD, SWaT: $+5.10\%$ USAD, HAI: $+3.81\%$ GDN) and NDCG@150% (SMD: $+27.97\%$ TranAD, SWaT: $+4.76\%$ GDN, HAI: $+2.93\%$ GDN). SARAD also outperforms on the range-based IPS@150% on most datasets (SMD: $43.19\%$ TranAD, SWaT: $33.43\%$ TranAD). Unlike SMD which performs forensic diagnosis to label anomalous features, SWaT and HAI label only the origins of cyberattacks as diagnosis labels. Consequentially, attack origins sometimes might not behave anomalously, e.g., attacks had failed, or the full set of anomalous features were not identified, thus diminishing the performance numbers on SWaT and HAI. SARAD generally outperforms detectors underpinned by temporal modeling due to its sensitivity to spatial associative changes. SARAD also outperforms spatial detectors such as GDN whose prediction errors limit its temporal scope to a single time point. Standard deviations of Table 4 are reported in Appendix L.

**Visualization** Figure 3 visualizes a real-world anomaly example via SARAD. Our detector captures the significant SAR caused by the anomalous features. The loose reconstruction of the progression raises the progression-based score $p$ and, in turn, the joint detection score $s$. Taking a broader view of the series in Fig. 3h, SAR significantly raises the scores at the start of the anomaly, even when the data-based errors $r$ are small. SARAD exploits SAR to achieve more robust anomaly detection.

Table 4: Anomaly diagnosis performance. Point-based HR@$P$%, NDCG@$P$%, and range-based IPS@$P$% are reported. All values are average percentages from five random seeds.

| Method | SMD HR@$P$% | | ND@$P$% | | IPS@$P$% | | SWaT HR@$P$% | | ND@$P$% | | IPS@$P$% | | HAI HR@$P$% | | ND@$P$% | | IPS@$P$% | |
|---|---|---|---|---|---|---|---|---|---|---|---|---|---|---|---|---|---|---|
| $P$ | 100 | 150 | 100 | 150 | 100 | 150 | 100 | 150 | 100 | 150 | 100 | 150 | 100 | 150 | 100 | 150 | 100 | 150 |
| TranAD | 33.33 | 45.26 | 34.71 | 41.81 | 22.21 | 33.25 | 4.82 | 6.36 | 4.82 | 5.76 | 17.47 | 20.91 | 4.08 | 6.27 | 3.98 | 5.33 | **12.67** | **21.78** |
| ATF-UAD | 27.80 | 40.94 | 26.67 | 34.46 | 17.34 | 26.16 | 1.85 | 3.19 | 1.83 | 2.65 | 5.45 | 8.28 | 2.96 | 5.22 | 3.08 | 4.48 | 5.04 | 8.59 |
| USAD | 27.03 | 39.74 | 25.83 | 33.44 | 18.31 | 26.07 | 4.39 | 7.73 | 4.39 | 6.48 | 12.83 | 27.17 | 3.78 | 5.83 | 3.76 | 5.01 | 12.37 | 16.22 |
| GDN | 28.67 | 41.57 | 28.62 | 36.27 | 21.18 | 30.76 | 5.99 | 7.21 | 6.13 | 6.89 | 14.04 | 21.72 | 5.45 | 8.29 | 5.50 | 7.23 | 7.41 | 12.07 |
| DiffAD | N/A | N/A | N/A | N/A | N/A | N/A | 1.82 | 2.90 | 1.81 | 2.47 | 5.45 | 12.63 | 3.32 | 4.79 | 3.41 | 4.32 | 12.00 | 17.26 |
| **Ours** | **56.73** | **71.93** | **60.79** | **69.78** | **61.38** | **76.44** | **9.57** | **12.83** | **9.61** | **11.65** | **35.45** | **54.34** | **6.45** | **12.10** | **6.69** | **10.16** | 7.48 | 14.07 |

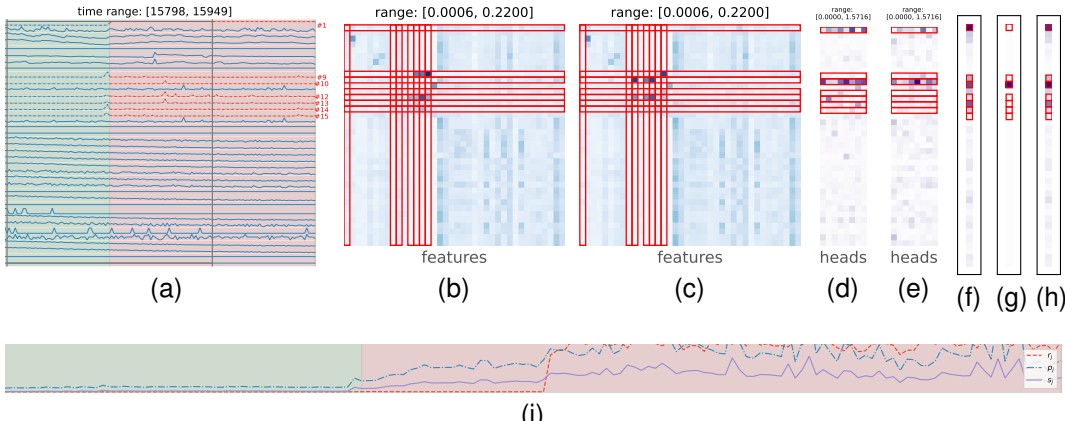

Figure 3: Visualization of applying SARAD for detection on SMD. 3a shows the raw time series right before and during an anomaly $p_i$ (colored in red). An input time window for SARAD is bounded in the black box. 3b and 3c show the average association mapping $\bar{\boldsymbol{A}}^L$ via final $L$-th layer's MHSA. 3d shows the aggregated progression $\boldsymbol{S}$ according to Eq. 3. 3e is its reconstruction. 3f, 3g, 3h show the scores $p$, $r$, and joint $s$ according to Eq. 5 per feature. 3i shows the anomaly scores for 3a's segment. Anomalous features (#1, #9, #10, #12, #13, #14 ,and #15) are highlighted with red bounding boxes.

**Complexity and Time Overheads** SARAD incurs 32 mins for training and and 0.39 ms for inference per sample on HAI, the largest dataset, falling far below the data collection time and sampling frequency. Those numbers are comparable with baselines and detailed in Appendix N.

### 4.3 Ablation Studies

**Spatial Progression Reconstruction** To evaluate the effectiveness of the progression module, we perform ablation studies on its submodules in Table 5. Standard deviations are reported in Appendix K. Removing the ReLU, i.e., to capture both association increases and reductions, in progression loses the focus on asscoation reduction and impairs the detection performance. Replacing the column sum operation in aggregation with the row sum which characterizes the disconnection of anomalous features from others and is shown to be less effectiveness than the column sum representing the drop out rates. Fully concatenating without sum operation dilutes the reduction patterns and significantly hurts the detection, at a cost of complexity. For the detection submodule, using the progression directly instead of the reconstruction errors registers reductions as anomalies directly and underperforms except on SMD due to its inability to rule out normal reduction patterns.

**Choice of Detection Criterion** Table 6 compares the detection performance using Eq. 5 (Joint), using only data-based $r$ (DR), and using only progression-based $p$ (SPR). While the data reconstruction is a robust criterion of anomalousness, SARAD embeds the spatial information into the joint criterion and outperforms either single criterion overall. Standard deviations are reported in Appendix K.

Additional ablation studies on the choice of diagnosis criterion are detailed in Appendix D.

Table 5: Anomaly detection performance under progression reconstruction changes.

| Submodule | Change | SMD | | | | PSM | | | | SWaT | | | | HAI | | | |
|---|---|---|---|---|---|---|---|---|---|---|---|---|---|---|---|---|---|
| | | $A_{ROC}$ | $A_{PR}$ | $V_{ROC}$ | $V_{PR}$ | $A_{ROC}$ | $A_{PR}$ | $V_{ROC}$ | $V_{PR}$ | $A_{ROC}$ | $A_{PR}$ | $V_{ROC}$ | $V_{PR}$ | $A_{ROC}$ | $A_{PR}$ | $V_{ROC}$ | $V_{PR}$ |
| **Ours** | - | 79.97 | 15.09 | 79.67 | 15.02 | 61.87 | 41.06 | 61.77 | 41.01 | **88.29** | **72.90** | **87.52** | **70.68** | **96.87** | **67.78** | **96.17** | **64.70** |
| Prog. (Eq.2) | no ReLU | 75.43 | 12.38 | 75.16 | 12.35 | 60.35 | 40.32 | 60.21 | 40.24 | 87.48 | 67.88 | 86.75 | 66.04 | 95.59 | 64.81 | 94.93 | 61.87 |
| Aggr. (Eq.3) | row sum | 79.47 | 15.87 | 79.17 | 15.82 | **63.36** | **42.41** | **63.20** | **42.33** | 88.19 | 70.12 | 87.40 | 68.12 | 96.56 | 67.26 | 95.80 | 64.21 |
| Aggr. (Eq.3) | no sum | 57.31 | 6.09 | 56.98 | 6.08 | 47.50 | 33.92 | 47.34 | 33.86 | 86.10 | 69.69 | 85.26 | 67.71 | 91.63 | 57.49 | 90.70 | 55.01 |
| Detection | $\mathcal{S}$ directly | **80.42** | **15.79** | **80.14** | **15.73** | 60.59 | 39.71 | 60.49 | 39.63 | 87.33 | 69.38 | 86.70 | 67.59 | 95.72 | 64.95 | 94.98 | 61.91 |

Table 6: Anomaly detection performance under different choices of detection criterion.

| Method | Criter. | SMD | | | | PSM | | | | SWaT | | | | HAI | | | |
|---|---|---|---|---|---|---|---|---|---|---|---|---|---|---|---|---|---|
| | | $A_{ROC}$ | $A_{PR}$ | $V_{ROC}$ | $V_{PR}$ | $A_{ROC}$ | $A_{PR}$ | $V_{ROC}$ | $V_{PR}$ | $A_{ROC}$ | $A_{PR}$ | $V_{ROC}$ | $V_{PR}$ | $A_{ROC}$ | $A_{PR}$ | $V_{ROC}$ | $V_{PR}$ |
| **Ours** | both | **79.97** | 15.09 | **79.67** | 15.02 | 61.87 | 41.06 | 61.77 | 41.01 | **88.29** | **72.90** | **87.52** | **70.68** | **96.87** | **67.78** | **96.17** | **64.70** |
| DR | $r$ only | 73.21 | 13.53 | 72.72 | 13.44 | 61.53 | 39.99 | 61.45 | 39.93 | 86.76 | 68.66 | 86.15 | 67.03 | 96.28 | 66.38 | 95.56 | 63.33 |
| SPR | $p$ only | 79.71 | **15.44** | 79.39 | **15.36** | **62.41** | **41.47** | **62.20** | **41.36** | 65.87 | 20.10 | 65.31 | 19.53 | 95.54 | 61.73 | 94.77 | 59.07 |

## 5  Conclusion

In this work, we propose SARAD for time series anomaly detection and diagnosis. The approach effectively exploits the spatial association descending patterns of anomalies. Data reconstruction with Transformer guides learning of spatial associations from data and captured as progression, while progression reconstruction quantifies the anomalous association descent and complements the insensitivity of the former to spatial disassociation during anomalies. SARAD experimentally demonstrates state-of-the-art detection and diagnosis performance and foreshadows the power of spatial modeling for related time series tasks.

## Acknowledgments and Disclosure of Funding

Calculations were performed using the Sulis Tier 2 HPC platform hosted by the Scientific Computing Research Technology Platform at the University of Warwick. Sulis is funded by EPSRC Grant EP/T022108/1 and the HPC Midlands+ consortium. The authors acknowledge the use of the Batch Compute System in Department of Computer Science at the University of Warwick, and associated support services, in the completion of this work.

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

# A  Broader Impacts

The broader impact of the work presented rests in the increasingly pervasive nature of industrial control systems, intrusion detection systems, and remote monitoring solutions in healthcare contexts, all of which commonly utilize some form of anomaly detection. Further to the time series analysis that is commonplace, the work presented in this paper demonstrates how Transformer can be used to learn the spatial associations that ubiquitously characterize these feedback-controlled systems, supplementing time series analysis to provide state-of-the-art performance. As such, the work presented has broad applicability, whilst explicitly targeting automated industrial control systems.

# B  Limitations

While the model size scales linearly with the number of features, the time complexity of SARAD is quadratic with respect to the features. SARAD could incur significant training and inference overheads when the supervisory system is extensively large. We caution that the overheads of the largest dataset in our experiments fall well below data collection overhead and sampling frequency (see Appendix N). To scale, we will explore hierarchical time series anomaly detection via clustering. Another limitation of this work is the scarcity of forensically labeled datasets like SMD for anomaly diagnosis, not least due to the intensive labor and domain knowledge implied. To address that gap, we will explore publicly available audit and operational time series data for sources.

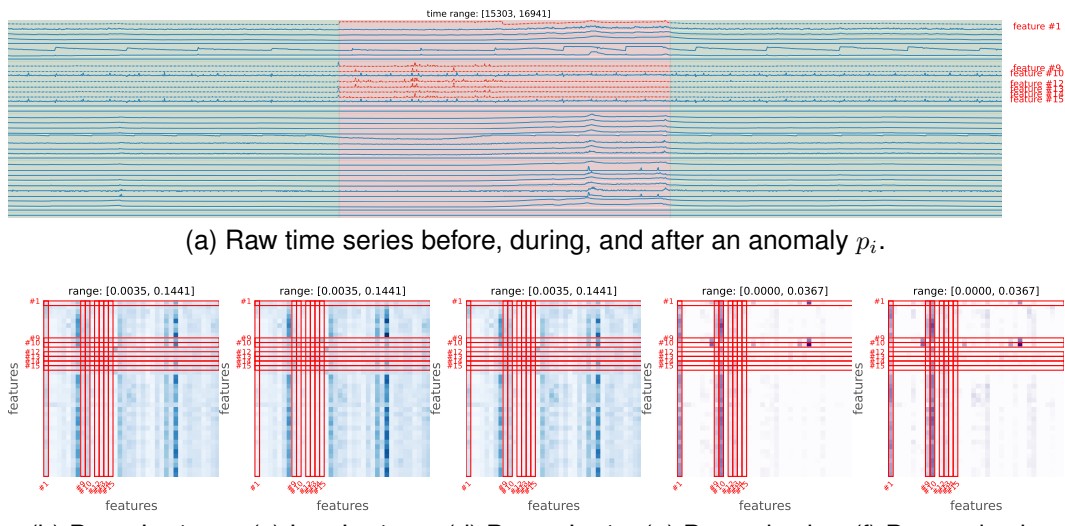

(a) Raw time series before, during, and after an anomaly $p_i$.

(b) Pre-$p_i$'s $\boldsymbol{A}$. (c) In-$p_i$'s $\boldsymbol{A}$. (d) Post-$p_i$'s $\boldsymbol{A}$. (e) Pre-reduction. (f) Post-reduction.

Figure 4: Spatial associations captured by Transformer on SMD (Su et al., 2019b).

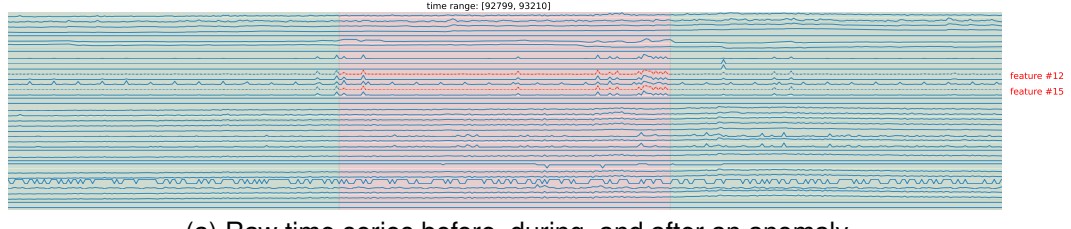

(a) Raw time series before, during, and after an anomaly $p_i$.

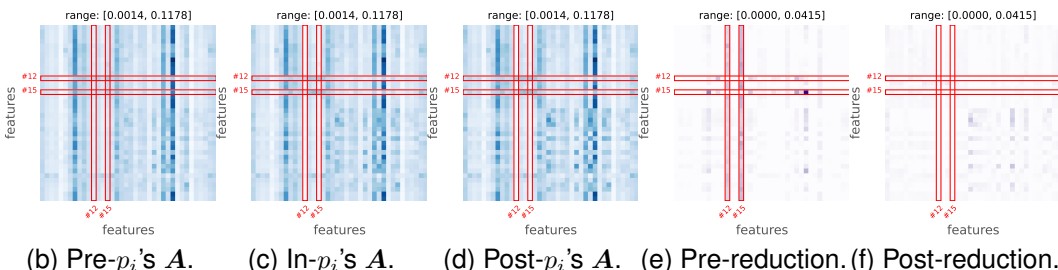

(b) Pre-$p_i$'s $\boldsymbol{A}$. (c) In-$p_i$'s $\boldsymbol{A}$. (d) Post-$p_i$'s $\boldsymbol{A}$. (e) Pre-reduction. (f) Post-reduction.

Figure 5: Spatial associations captured by Transformer on SMD (Su et al., 2019b).

## C   Examples of Spatial Association Reduction

As mentioned in Section 1, herein we provide more real-world examples of Spatial Association Reduction (SAR) exhibited by time series anomalies. Figures 4, 5, 6, 7 showcase the spatial associations captured within Transformer via MHSA. Subfigures in each aforementioned figure show, in that order, (a) the raw time series right before, during, and after an anomaly $p_i \in \mathcal{P}$ (colored in red), (b) association mapping $\boldsymbol{A}_h^L$ output by final $L$-th layer's MHSA are averaged across heads to derive average association $\boldsymbol{A}$ before $p_i$, (c) $\boldsymbol{A}_h^L$ during $p_i$, (d) $\boldsymbol{A}_h^L$ after $p_i$ wherein brighter cells have smaller values and anomalous features are highlighted with red bounding boxes, (e) the reduction-only changes from before the anomaly to during the anomaly, i.e., ReLU($\boldsymbol{A}_{\text{pre}} - \boldsymbol{A}_{\text{in}}$), and finally (f) the reduction-only changes from after the anomaly to during the anomaly, i.e., ReLU($\boldsymbol{A}_{\text{post}} - \boldsymbol{A}_{\text{in}}$).

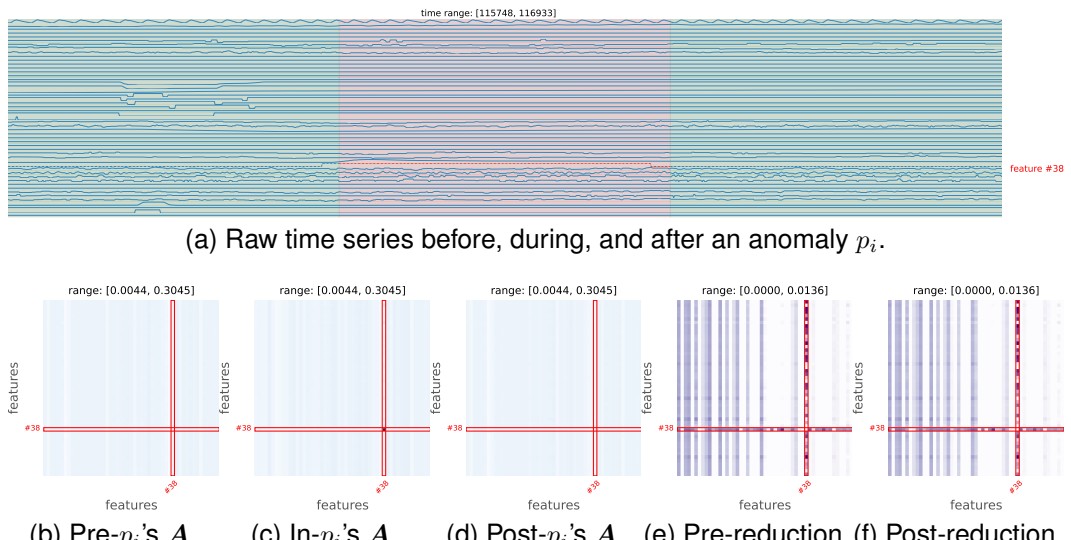

(a) Raw time series before, during, and after an anomaly $p_i$.

(b) Pre-$p_i$'s $\boldsymbol{A}$. (c) In-$p_i$'s $\boldsymbol{A}$. (d) Post-$p_i$'s $\boldsymbol{A}$. (e) Pre-reduction. (f) Post-reduction.

Figure 6: Spatial associations captured by Transformer on SWaT (Mathur and Tippenhauer, 2016).

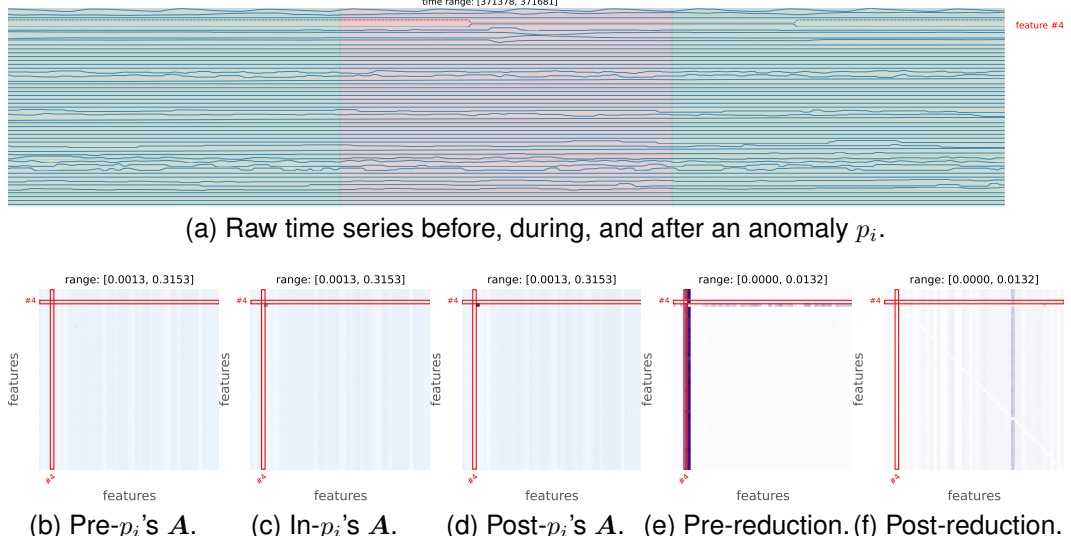

(a) Raw time series before, during, and after an anomaly $p_i$.

(b) Pre-$p_i$'s $\boldsymbol{A}$. (c) In-$p_i$'s $\boldsymbol{A}$. (d) Post-$p_i$'s $\boldsymbol{A}$. (e) Pre-reduction. (f) Post-reduction.

Figure 7: Spatial associations captured by Transformer on SWaT (Mathur and Tippenhauer, 2016).

Table 7: Anomaly diagnosis performance under different choices of diagnosis criterion.

| Method | Criter. | SMD HR@$P$% | | ND@$P$% | | IPS@$P$% | | SWaT HR@$P$% | | ND@$P$% | | IPS@$P$% | | HAI HR@$P$% | | ND@$P$% | | IPS@$P$% | |
|---|---|---|---|---|---|---|---|---|---|---|---|---|---|---|---|---|---|---|---|
| $P$ | | 100 | 150 | 100 | 150 | 100 | 150 | 100 | 150 | 100 | 150 | 100 | 150 | 100 | 150 | 100 | 150 | 100 | 150 |
| **Ours** | $r_j$ only | **56.73** | **71.93** | **60.79** | **69.78** | **61.38** | **76.44** | 9.57 | 12.83 | 9.61 | 11.65 | **35.45** | **54.34** | 6.45 | 12.10 | 6.69 | 10.16 | 7.48 | 14.07 |
| SPR | $p_j$ only | 42.97 | 57.33 | 46.32 | 54.85 | 52.01 | 64.82 | **14.32** | **17.18** | **14.40** | **16.18** | 20.10 | 37.37 | 5.71 | 8.50 | 5.86 | 7.58 | 9.70 | **17.04** |
| Joint | both | 48.91 | 61.49 | 53.12 | 60.59 | 56.56 | 70.06 | 2.60 | 3.50 | 2.66 | 3.20 | 16.16 | 20.40 | **10.28** | **15.42** | **10.74** | **13.92** | **12.59** | 16.52 |

# D   Choice of Diagnosis Criterion

Concerning the rationality of data-only diagnosis criterion in Eq. 6, we consider an alternate joint diagnosis criterion in line with the detection criterion in Eq. 5.

$$r_j = ||\hat{X}_{(j,\cdot)} - X_{(j,\cdot)}||_2^2, \; p_j = ||\hat{S}_{(\cdot,j)} - S_{(\cdot,j)}||_2^2, \; s_j = (r_j - \mu_{r_j})/\sigma_{r_j} + (p_j - \mu_{p_j})/\sigma_{p_j} \quad (7)$$

where $r_j$ is feature $j$'s data reconstruction error, $p_j$ its progression reconstruction error, $\mu_{r_j}, \mu_{p_j}$ the means of $r_j, p_j$ on the validation set, and $\sigma_{r_j}, \sigma_{p_j}$ the standard deviation of $r_j, p_j$. Table 7 compares the diagnosis performance using only $r_j$ (SARAD), using only $p_j$ (SPR), and using Eq. 7 (Joint). Unlike in anomaly detection, the great discrepancy between $r_j$ and $p_j$ more than often degrades the performance of the joint criterion. SARAD uses $r_j$ only which produces suboptimal and yet reliable performance in the longer anomalous horizons. Standard deviations are reported in Appendix L.

# E  Datasets

We include four real-world datasets collected under industrial control and IT service monitoring settings for evaluations. Anomalies range from IT service outages to external cyber-physical attacks against control systems.

## E.1  Dataset Descriptions

1. **Server Machine Dataset (SMD)** (Su et al., 2019b,a) is a server metric dataset from a large-scale IT company. Engineers annotated anomalous events in the second half of the data with indicator-level attributions.

2. **Pooled Server Metrics (PSM)** dataset (Abdulaal et al., 2021a,b) captures key performance indicators of servers on an online shopping platform. Website engineers annotated anomalous events for data in the last eight weeks.

3. **Secure Water Treatment (SWaT)** dataset (Mathur and Tippenhauer, 2016; iTrust, 2023) contains sensor readings and actuator status on a minuscule real-world water treatment system during a six-day normal operational period. A knowledgeable attacker performed 36 cyber-physical attacks during a five-day attack period and they are labelled as anomalous accordingly.

4. **HIL-based Augmented ICS (HAI)** dataset (Shin et al., 2021b,a) records measurements and control actions within a Hardware-In-the-Loop (HIL) dual power (steam-turbine and hydropower) generation testbed during its two-week operation. Both single-point primitive and multi-point combined attacks are performed on the testbed to emulate a threat actor with cyber-physical capacities. We use the 21.03 version of HAI.

All training sets contain only unlabeled data and the test sets contain data with anomaly labels. Statistics of the datasets are given in Table 2.

## E.2  Lengths of Anomalies

We further characterize the detection datasets by the lengths of the anomalous events. Figure 8 shows the empirical cumulative distribution function of the anomalous lengths. SWaT has the longest median length of 447 among the four datasets considered, followed by HAI (162), SMD (11), and lastly PSM (5). The very short lengths on SMD and PSM benefit temporal detectors which tend to embed a single or few time points (see Appendix M), whereas SARAD adopts a half time window embedding strategy. The catch is that SARAD can learn spatial relationships with temporal aggregated information per feature, while temporal detectors could not, bringing about benefits of performing anomaly detection in the spatial association space. A more scalable approach for temporal aggregation is to be explored in the future, though variable window sizes or subseries splits might be implied.

## E.3  Lengths of Diagnosis Labels

Figure 8 shows the empirical cumulative distribution function of the diagnosis label lengths. Whereas SMD forensically labels features which deviate from their normal behavioral patterns as anomalous, SWaT and HAI only label the points of attacks as anomalous as the their creators have advanced knowledge of such attacks. The latter labeling strategy results in incomplete sets of anomalous features and diminishes the diagnosis performance of all models, as evidence in Table 4 in Section 4.2.

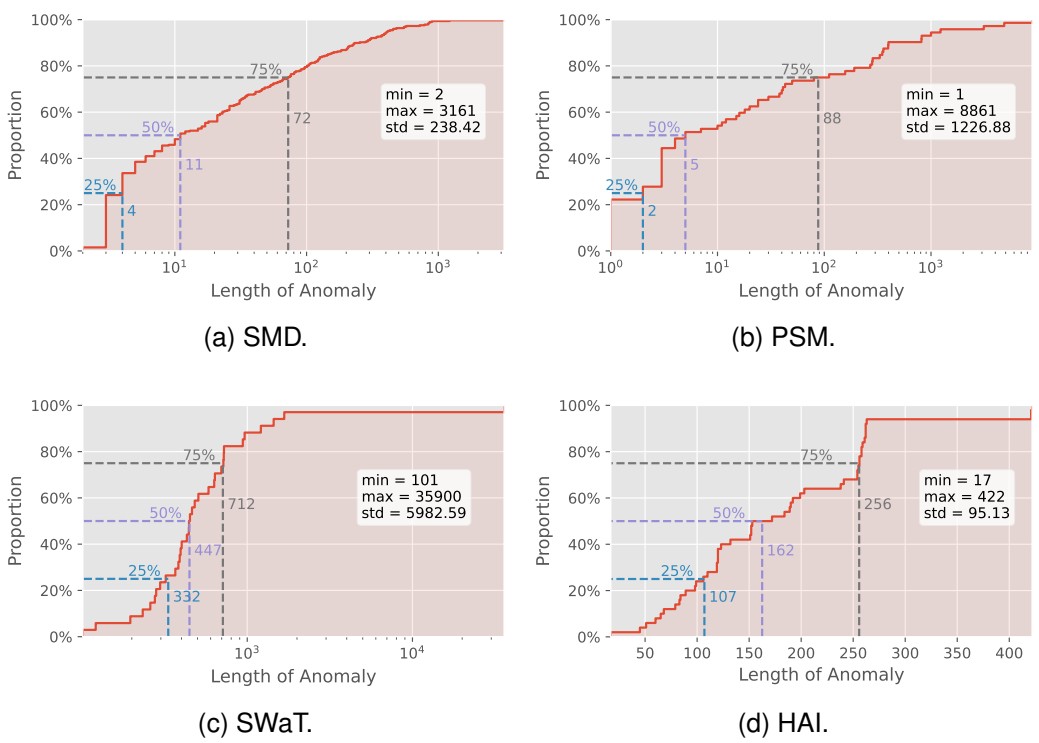

Figure 8: Empirical distribution function of the lengths of anomalies.

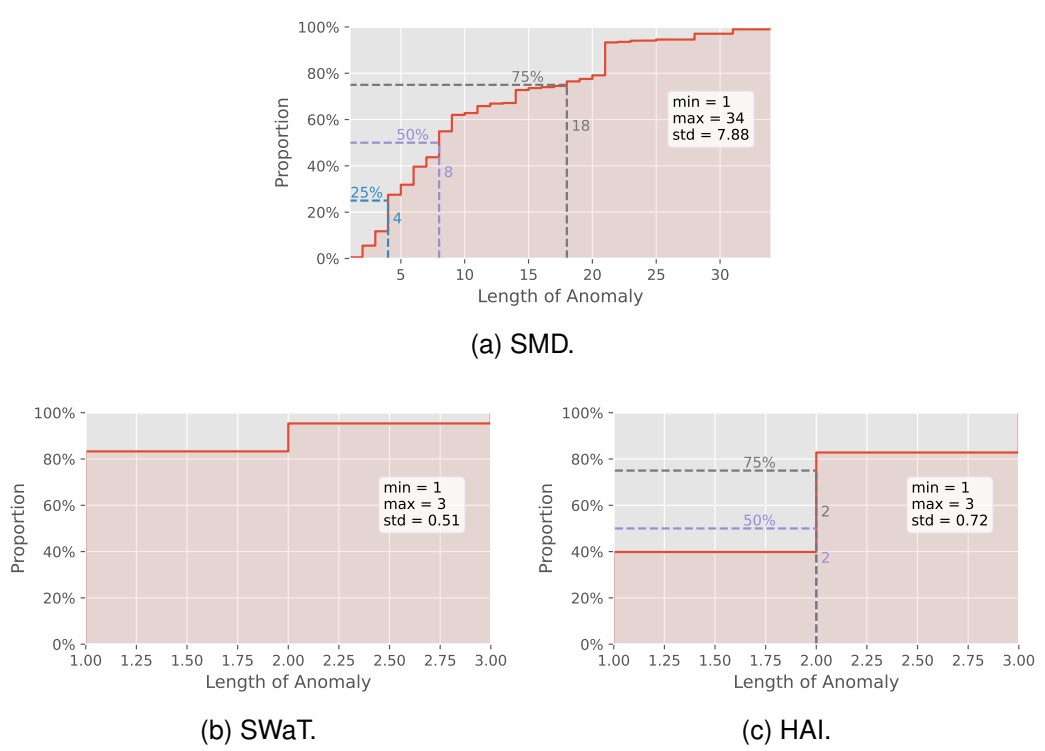

Figure 9: Empirical distribution function of the lengths of diagnosis labels.

# F  Implementation Details

We implement SARAD in Python using pyTorch library (Paszke et al., 2019) and Hydra framework (Yadan, 2019). All experiments are run on a single NVIDIA A10 (24GB) GPU. Adam optimizer (Kingma and Ba, 2015) is used and learning rate is halved every epoch for 3 epochs to prevent over-fitting. The time window size is $2W = 100$. The data reconstruction module has $H = 8$ attention heads per layer with attention length $D = 512$ and hidden length $D_{FF} = 2048$. For hyperparameter tuning, training set is temporally partitioned into 80% for training and 20% for validation. On each dataset we first perform TPE sampling (Bergstra et al., 2011) for number of encoding layers $L \in \{3, 5\}$ and learning rate $\in [10^{-4}, 10^{-2}]$ to derive the best data reconstruction loss $L_R$ on the validation set. The progression module by default has hidden length of $D_P = 64$. We then perform TPE sampling to search weight $\lambda_{L_S} \in [10^{-2}, 10^2]$ for the progression reconstruction loss $L_S$ on the validation set.

# G  Open-accessed Code and Data

During the review period, code is anonymized and openly available at `https://github.com/daidahao/SARAD/` with specific instructions and scripts to reproduce experimental results. All data used in our experiments can be openly accessed from public repositories or requested via original authors' websites. Full links are provided in Appendix E.

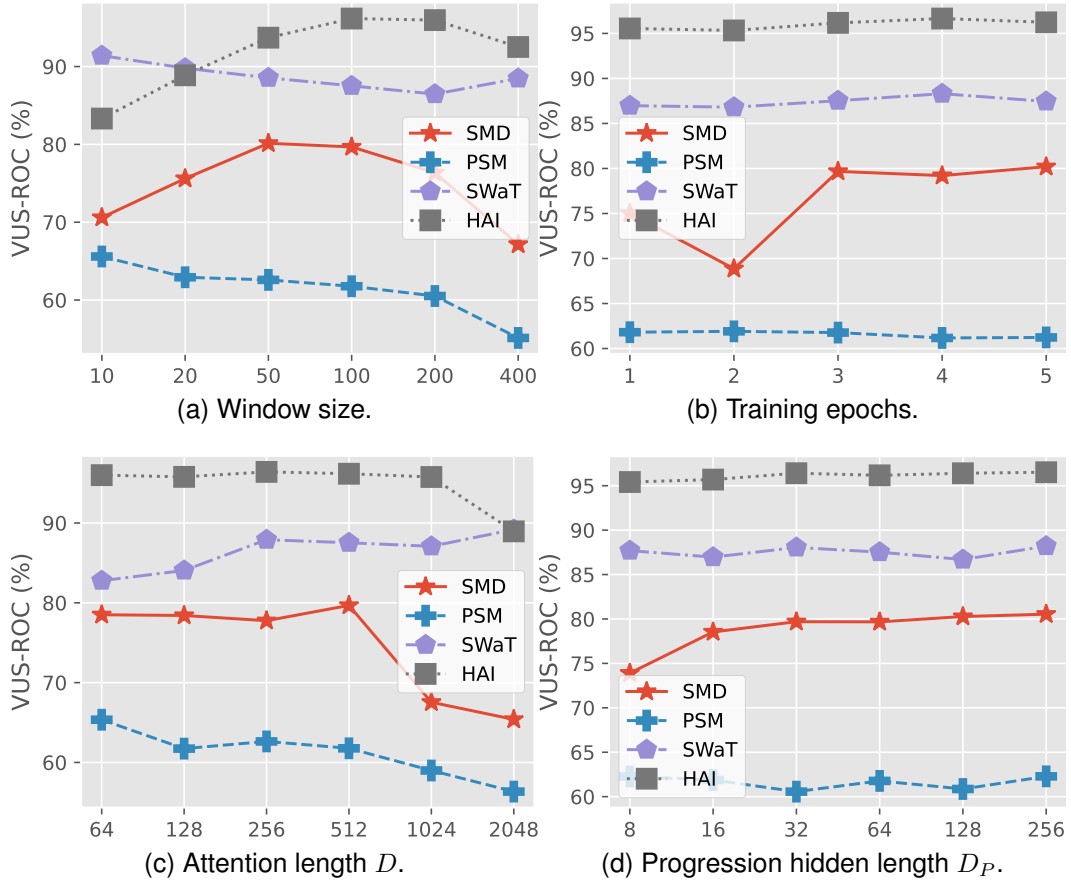

Figure 10: Hyperparameter Sensitivity of detection performance in VUS-ROC scores.

## H   Hyperparameter Sensitivity

We examine the hyperparameter sensitivity of SARAD's detection performance. Concretely, we consider the effects of the sliding window size (default is 100), the number of training epochs (3), the attention length of Transformer encoding layers $D$ (512), and the hidden length of the progression reconstruction module $D_P$ (64). Figure 10 and 11 present the results. On the window size, a sliding window too small confines temporal modeling to small temporal receptive fields and contains anomaly detection in the association space. However, a sliding window too large incurs higher computational costs, although unlike temporal modeling the costs here are linear. On datasets with shorter anomalous lengths such as SMD and PSM, the anomalous patterns are diluted even further, resulting in performance degradation.

On the number of training epochs, fewer epochs lead to model underfitting, and yet overfitting is largely prevented with more epochs due to the aggressive learning rate halving per epoch. On the attention length $D$, a larger Transformer is prone to overfitting with visible performance drops on SMD and PSM as $D$ passes the default 512, both of whose monitored systems are smaller in scale. On the hidden length $D_P$, a more complex progression anomaly detector does not adversely impact the performance, suggesting that the association space is less prone to detection overfitting than the data space.

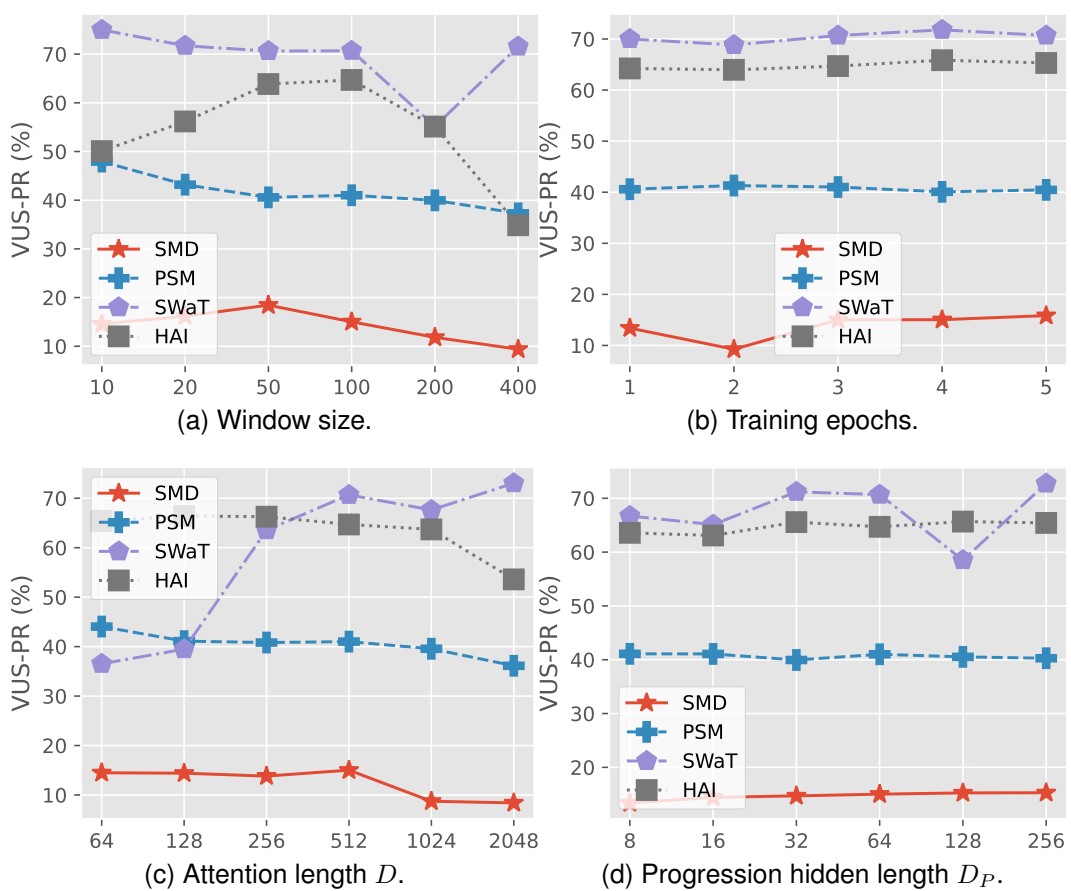

Figure 11: Hyperparameter Sensitivity of detection performance in VUS-PR scores.

# I Detection Metrics

Conventional metrics such as precision, recall, F1 and the threshold-independent Area Under the Curve (AUC) scores are commonly point-based, i.e., predicted labels are scored individually by time points (Zhou et al., 2023; Zhang et al., 2022; Xu et al., 2018). Real-world benchmarks are rife with range-wise anomalies spanning consecutive time points (Wagner et al., 2023). Point-based metrics are generally ill-suited for evaluating detection performance due to the continuous-discrete conversion and the series-label misalignment (labeling anomalies precisely is hard) (Garg et al., 2022; Tatbul et al., 2018). Here, we use the range-based metrics proposed in (Paparrizos et al., 2022). Compared against their point-based counterparts, they provide robustness to labeling delay and scoring noises as well as performant detector separability and series consistency.

Given a set of anomalous ranges $\mathcal{P} = \{p_i = (s_i, e_i)\}$ wherein each anomaly $p_i$ starts at timestamp $s_i$ and ends at $e_i$, we enclose each range with uniform $l/2$-length preceding and succeeding buffers. Given the anomaly label $y_t \in \{0, 1\}$ at each timestamp $t$, we derive a new soft label $\tilde{y}_t \in [0, 1]$ as per the minimum temporal distance of $t$ to any anomaly $p_i \in \mathcal{P}$:

$$\tilde{y}_t = \begin{cases} \sqrt{1 - |s_i - t|/l}, & \exists p_i \in \mathcal{P}, t \in [s_i - l/2, s_i) \\ \sqrt{1 - |t - e_i|/l}, & \exists p_i \in \mathcal{P}, t \in (e_i, e_i + l/2] \\ y_t, & \text{otherwise} \end{cases} \quad (8)$$

where $l$ is the buffer length, normally set to the median segment length in $\mathcal{P}$. Within the buffers, $\tilde{y}_t$ monotonically increases from $\sqrt{2}/2$ to 1 as the distance decreases. With the new soft label series $\tilde{\mathcal{Y}} = \{\tilde{y}_t\}$, we define the True Positives (TP), False Positives (FP), True Negatives (TN), and False Negatives (FN) accordingly:

$$TP = \tilde{\mathcal{Y}}^T \cdot \hat{y}, \ FP = (1 - \tilde{\mathcal{Y}})^T \cdot \hat{y}, \ TN = (1 - \tilde{\mathcal{Y}})^T \cdot (1 - \hat{y}), \ FN = \tilde{\mathcal{Y}}^T \cdot (1 - \hat{y}) \quad (9)$$

We then compute the threshold-independent AUC for the Receiver Operating Characteristic (AUC-ROC), i.e., TP rate vs FP rate, and the Precision-Recall (AUC-PR) curves respectively to be rid of thresholding impact. Fully parameter-free Volume Under the Surface (VUS) scores for AUC-ROC (VUS-ROC) and AUC-PR (VUS-PR) are also computed under different buffer lengths $\hat{l} \in [0, 2l]$.

## J  Diagnosis Metrics

In line with previous works (Tuli et al., 2022; Zhao et al., 2020), we use common metrics such as Hit Rate (HR) (Su et al., 2019b) and Normalized Discounted Cumulative Gain (NDCG) (Järvelin and Kekäläinen, 2002) to measure performance where diagnosis labels are available. Given a set of anomalous features $G_i \subseteq [N]$ at an anomalous timestamp $t \in p_i \in \mathcal{P}$ as a diagnosis ground-truth and the set of top $k$-ranked features $\Gamma_t@P\%$ according to Eq. 5 where $k = \lceil |G_i| \times P\% \rceil$, say $k = 5$ when $|G_i| = 3$ and $P = 150$, the HR at $P\%$ ($P \geq 100$) features is the overlap ratio between the two:

$$\mathrm{HR}_t@P\% = \frac{|G_i \cap \Gamma_t@P\%|}{|G_i|} \tag{10}$$

In information retrieval, DCG measures the cumulative utility of retrieved documents by their ranking order up to a certain position. NDCG normalizes the DCG by the maximum possible DCG. They are parameterized by $P\%$ to determine the location in our evaluation and calculated as follows.

$$\mathrm{DCG}_t@P\% = \sum_{j=1}^{k} \frac{r_j}{\log_2(j+1)}, \ \mathrm{IDCG}_t = \sum_{j=1}^{|G_i|} \frac{1}{\log_2(j+1)}, \ \mathrm{NDCG}_t@P\% = \frac{\mathrm{DCG}_t@P\%}{\mathrm{IDCG}_t} \tag{11}$$

where $r_j \in \{0, 1\}$ is the relevance value of the $j$-th element and, in this case, the membership of $\Gamma_t@P\%$'s $j$-th feature in $G_i$. NDCG has a value strictly between 0 and 1.

At the range level, we measure the **Interp**retation Score (IPS) initially proposed in Li et al. (2021) and here expanded to fit the $P\%$ parameterization. For each anomalous range $p_i$, the IPS score is:

$$\mathrm{IPS}_i@P\% = \frac{|G_i \cap \Omega_i@P\%|}{|G_i|} \tag{12}$$

where $\Omega_i@P\%$ is the top $k$-ranked features according to $\max_{t \in p_i} s_{j,t}$, the $j$-th feature's maximum anomaly score during $p_i$ and $k = \lceil |G_i| \times P\% \rceil$. It is the HR equivalence at the range level as per the highest anomalous scores per feature.

Table 8: Standard deviations of anomaly detection performance in Table 3. Standard deviations of threshold-independent AUC-ROC and AUC-PR metrics and fully parameter-free VUS-ROC and VUS-PR metrics are reported. All values are percentages.

| Method | SMD | | | | PSM | | | | SWaT | | | | HAI | | | |
|---|---|---|---|---|---|---|---|---|---|---|---|---|---|---|---|---|
| | $A_{ROC}$ | $A_{PR}$ | $V_{ROC}$ | $V_{PR}$ | $A_{ROC}$ | $A_{PR}$ | $V_{ROC}$ | $V_{PR}$ | $A_{ROC}$ | $A_{PR}$ | $V_{ROC}$ | $V_{PR}$ | $A_{ROC}$ | $A_{PR}$ | $V_{ROC}$ | $V_{PR}$ |
| IF | 0.45 | 0.11 | 0.40 | 0.11 | 1.15 | 1.11 | 1.13 | 1.10 | 1.42 | 2.49 | 1.56 | 2.58 | 1.30 | 0.94 | 1.25 | 0.79 |
| DIF | 0.81 | 0.16 | 0.76 | 0.15 | 0.94 | 0.64 | 1.00 | 0.64 | 0.26 | 0.73 | 0.33 | 0.90 | 1.65 | 3.31 | 1.64 | 3.18 |
| TranAD | 0.21 | 0.11 | 0.23 | 0.10 | 0.66 | 0.29 | 0.62 | 0.28 | 12.15 | 5.30 | 11.92 | 5.30 | 8.89 | 9.83 | 8.83 | 9.61 |
| ATF-UAD | 2.23 | 0.12 | 2.19 | 0.12 | 4.19 | 1.75 | 4.33 | 1.74 | 12.06 | 6.39 | 11.92 | 6.27 | 10.83 | 12.11 | 10.77 | 11.86 |
| AT | 0.39 | 0.50 | 0.39 | 0.48 | 3.05 | 0.87 | 3.22 | 0.86 | 3.51 | 1.43 | 3.61 | 1.43 | 5.50 | 1.96 | 5.24 | 1.90 |
| DCdetector | 0.39 | 0.12 | 0.39 | 0.12 | 1.72 | 0.13 | 1.57 | 0.13 | 0.06 | 0.23 | 0.06 | 0.24 | N/A | N/A | N/A | N/A |
| USAD | 1.43 | 0.48 | 1.44 | 0.48 | 0.40 | 0.21 | 0.40 | 0.21 | 6.41 | 7.42 | 6.43 | 6.77 | 7.41 | 8.02 | 7.17 | 7.79 |
| GDN | 0.81 | 1.56 | 0.78 | 1.55 | 2.58 | 1.26 | 2.39 | 1.20 | 1.35 | 3.21 | 1.17 | 3.14 | 1.63 | 4.62 | 1.66 | 4.46 |
| MAD-GAN | 2.00 | 0.50 | 1.99 | 0.50 | 4.65 | 4.08 | 4.71 | 4.09 | 4.15 | 4.09 | 4.17 | 4.13 | 2.05 | 1.76 | 1.99 | 1.72 |
| DiffAD | 0.40 | 0.17 | 0.31 | 0.18 | 1.15 | 0.42 | 0.89 | 0.40 | 0.26 | 0.09 | 0.30 | 0.12 | 0.60 | 0.47 | 0.56 | 0.47 |
| **Ours** | 0.98 | 0.75 | 1.01 | 0.76 | 1.07 | 0.69 | 1.09 | 0.69 | 0.70 | 1.34 | 0.60 | 1.05 | 0.47 | 1.08 | 0.56 | 1.11 |

Table 9: Standard deviations of anomaly detection performance in Table 5. Standard deviations of threshold-independent AUC-ROC and AUC-PR metrics and fully parameter-free VUS-ROC and VUS-PR metrics are reported. All values are percentages.

| Submodule | Change | SMD | | | | PSM | | | | SWaT | | | | HAI | | | |
|---|---|---|---|---|---|---|---|---|---|---|---|---|---|---|---|---|---|
| | | $A_{ROC}$ | $A_{PR}$ | $V_{ROC}$ | $V_{PR}$ | $A_{ROC}$ | $A_{PR}$ | $V_{ROC}$ | $V_{PR}$ | $A_{ROC}$ | $A_{PR}$ | $V_{ROC}$ | $V_{PR}$ | $A_{ROC}$ | $A_{PR}$ | $V_{ROC}$ | $V_{PR}$ |
| **Ours** | - | 0.98 | 0.75 | 1.01 | 0.76 | 1.07 | 0.69 | 1.09 | 0.69 | 0.70 | 1.34 | 0.60 | 1.05 | 0.47 | 1.08 | 0.56 | 1.11 |
| Prog. (Eq.2) | no ReLU | 1.00 | 0.45 | 1.00 | 0.45 | 0.76 | 0.61 | 0.73 | 0.61 | 0.52 | 5.17 | 0.51 | 4.87 | 0.19 | 0.33 | 0.17 | 0.33 |
| Aggr. (Eq.3) | row sum | 0.87 | 0.61 | 0.88 | 0.62 | 1.30 | 1.30 | 1.29 | 1.28 | 1.39 | 3.62 | 1.40 | 3.61 | 0.22 | 0.67 | 0.21 | 0.60 |
| Aggr. (Eq.3) | no sum | 6.34 | 1.21 | 6.37 | 1.20 | 0.71 | 0.21 | 0.69 | 0.21 | 0.48 | 0.98 | 0.48 | 1.00 | 0.51 | 0.42 | 0.56 | 0.39 |
| Detection | $\mathcal{S}$ directly | 0.70 | 0.51 | 0.69 | 0.50 | 0.85 | 0.42 | 0.85 | 0.42 | 1.17 | 1.20 | 1.12 | 1.12 | 1.74 | 4.39 | 1.81 | 4.18 |

# K   Standard Deviations of Detection Performance

Table 8 reports the standard deviations of anomaly detetion performance as reported in Table 3 in Section 4.2.

Table 9 reports the standard deviations of anomaly diagnosis performance as reported in Table 5 in Section 4.3.

Table 10 reports the standard deviations of anomaly detection performance as reported in Table 6 in Section 4.3.

Table 10: Standard deviations of anomaly detection performance in Table 3. Standard deviations of threshold-independent AUC-ROC and AUC-PR metrics and fully parameter-free VUS-ROC and VUS-PR metrics are reported. All values are percentages.

| Method | Criter. | SMD | | | | PSM | | | | SWaT | | | | HAI | | | |
|---|---|---|---|---|---|---|---|---|---|---|---|---|---|---|---|---|---|
| | | $A_{ROC}$ | $A_{PR}$ | $V_{ROC}$ | $V_{PR}$ | $A_{ROC}$ | $A_{PR}$ | $V_{ROC}$ | $V_{PR}$ | $A_{ROC}$ | $A_{PR}$ | $V_{ROC}$ | $V_{PR}$ | $A_{ROC}$ | $A_{PR}$ | $V_{ROC}$ | $V_{PR}$ |
| **Ours** | both | 0.98 | 0.75 | 1.01 | 0.76 | 1.07 | 0.69 | 1.09 | 0.69 | 0.70 | 1.34 | 0.60 | 1.05 | 0.47 | 1.08 | 0.56 | 1.11 |
| DR | $r$ only | 0.82 | 0.19 | 0.56 | 0.15 | 1.76 | 1.00 | 1.76 | 1.01 | 1.07 | 1.52 | 1.11 | 1.72 | 0.44 | 0.86 | 0.46 | 0.81 |
| SPR | $p$ only | 0.99 | 0.83 | 1.02 | 0.85 | 0.39 | 0.18 | 0.42 | 0.20 | 1.61 | 0.67 | 1.64 | 0.63 | 1.10 | 3.30 | 1.21 | 3.32 |

Table 11: Standard deviations of anomaly diagnosis performance in Table 4. Standard deviations of point-based HR@$P$%, NDCG@$P$%, and range-based IPS@$P$% are reported. All values are percentages.

| Method | SMD | | | | | | SWaT | | | | | | HAI | | | | | |
| | HR@$P$% | | ND@$P$% | | IPS@$P$% | | HR@$P$% | | ND@$P$% | | IPS@$P$% | | HR@$P$% | | ND@$P$% | | IPS@$P$% | |
| $P$ | 100 | 150 | 100 | 150 | 100 | 150 | 100 | 150 | 100 | 150 | 100 | 150 | 100 | 150 | 100 | 150 | 100 | 150 |
| TranAD | 0.08 | 0.29 | 0.22 | 0.20 | 0.25 | 0.69 | 1.10 | 2.50 | 1.13 | 1.99 | 4.25 | 7.37 | 0.84 | 2.16 | 0.54 | 1.36 | 1.47 | 3.07 |
| ATF-UAD | 1.62 | 1.68 | 2.60 | 2.39 | 2.67 | 2.73 | 1.19 | 2.45 | 1.17 | 1.92 | 4.42 | 5.81 | 2.37 | 4.39 | 2.68 | 3.93 | 5.06 | 6.52 |
| USAD | 0.37 | 0.81 | 0.90 | 1.06 | 0.50 | 0.77 | 1.73 | 1.54 | 1.75 | 1.59 | 5.77 | 3.06 | 0.52 | 0.27 | 0.72 | 0.56 | 2.04 | 1.98 |
| GDN | 1.41 | 1.27 | 1.63 | 1.57 | 1.08 | 1.20 | 1.36 | 1.33 | 1.36 | 1.32 | 2.51 | 3.15 | 2.08 | 2.59 | 2.38 | 2.70 | 2.37 | 6.06 |
| DiffAD | N/A | N/A | N/A | N/A | N/A | N/A | 0.09 | 0.16 | 0.09 | 0.12 | 3.18 | 3.11 | 0.37 | 0.52 | 0.35 | 0.44 | 0.67 | 1.45 |
| **Ours** | 0.46 | 0.46 | 0.56 | 0.58 | 1.61 | 0.96 | 0.98 | 1.68 | 1.03 | 1.45 | 4.20 | 2.52 | 0.64 | 0.96 | 0.74 | 0.88 | 1.71 | 1.57 |

Table 12: Standard deviations of anomaly diagnosis performance in Table 7. Standard deviations of point-based HR@$P$%, NDCG@$P$%, and range-based IPS@$P$% are reported. All values are percentages.

| Method | SMD | | | | | | SWaT | | | | | | HAI | | | | | |
| | HR@$P$% | | ND@$P$% | | IPS@$P$% | | HR@$P$% | | ND@$P$% | | IPS@$P$% | | HR@$P$% | | ND@$P$% | | IPS@$P$% | |
| $P$ | 100 | 150 | 100 | 150 | 100 | 150 | 100 | 150 | 100 | 150 | 100 | 150 | 100 | 150 | 100 | 150 | 100 | 150 |
| **Ours** | 0.46 | 0.46 | 0.56 | 0.58 | 1.61 | 0.96 | 0.98 | 1.68 | 1.03 | 1.45 | 4.20 | 2.52 | 0.64 | 0.96 | 0.74 | 0.88 | 1.71 | 1.57 |
| SPR | 1.38 | 1.58 | 1.28 | 1.36 | 2.05 | 2.55 | 21.26 | 24.13 | 21.31 | 23.10 | 8.55 | 10.41 | 0.95 | 1.35 | 0.99 | 1.20 | 2.94 | 4.36 |
| Joint | 1.63 | 1.27 | 1.60 | 1.39 | 1.73 | 2.46 | 0.20 | 0.92 | 0.16 | 0.62 | 2.83 | 5.36 | 0.63 | 1.20 | 0.65 | 0.94 | 1.72 | 1.94 |

# L   Standard Deviations of Diagnosis Diagnosis

Table 11 reports the standard deviations of anomaly diagnosis performance as reported in Table 4 in Section 4.2.

Table 12 reports the standard deviations of anomaly diagnosis performance as reported in Table 7 in Appendix D.

Table 13: Anomaly detection point-based performance. Threshold-dependent **P**recision, **R**ecall, F1 scores and threshold-independent AUC-ROC and AUC-PR scores are reported. All values are average percentages from five random seeds. The best values are in **bold** and the second best underlined.

| Method | SMD | | | | | PSM | | | | | SWaT | | | | | HAI | | | | |
|---|---|---|---|---|---|---|---|---|---|---|---|---|---|---|---|---|---|---|---|---|
| | P | R | F1 | $A_{ROC}$ | $A_{PR}$ | P | R | F1 | $A_{ROC}$ | $A_{PR}$ | P | R | F1 | $A_{ROC}$ | $A_{PR}$ | P | R | F1 | $A_{ROC}$ | $A_{PR}$ |
| IF | 15.22 | 18.53 | 16.61 | 66.26 | 12.67 | 39.12 | 76.75 | 51.50 | **71.00** | 45.61 | 99.54 | 59.06 | 74.13 | 84.25 | 72.86 | 12.87 | 22.00 | 16.12 | 70.68 | 8.20 |
| DIF | 29.62 | 21.67 | **24.87** | 69.20 | **16.47** | 34.17 | 93.14 | 49.98 | 65.00 | 40.68 | 96.79 | 61.24 | 75.01 | **87.37** | **75.63** | 62.67 | 39.75 | 48.59 | 79.71 | 30.62 |
| TranAD | 18.54 | 12.79 | 14.12 | 52.34 | 8.44 | 34.04 | 96.53 | 50.33 | 62.09 | **45.74** | 26.45 | 74.01 | 36.62 | 57.92 | 16.60 | 65.27 | 30.33 | 39.05 | 73.22 | 31.03 |
| ATF-UAD | 6.07 | 24.06 | 9.49 | 52.23 | 4.71 | 30.50 | 91.67 | 45.56 | 57.70 | 38.35 | 30.09 | 73.37 | 40.54 | 62.47 | 17.89 | 73.25 | 25.37 | 37.18 | 68.62 | 26.67 |
| AT | 4.16 | 100.00 | 7.98 | 49.97 | 4.55 | 27.73 | 100.00 | 43.42 | 45.79 | 26.21 | 12.02 | 100.00 | 21.46 | 42.43 | 10.23 | 6.86 | 8.88 | 7.60 | 35.82 | 4.40 |
| DCdetector | 4.20 | 100.00 | 8.05 | 49.12 | 4.19 | 25.01 | 100.00 | 40.01 | 49.47 | 24.82 | 11.82 | 100.00 | 21.14 | 49.93 | 11.82 | N/A | N/A | N/A | N/A | N/A |
| USAD | 14.15 | 22.20 | 17.12 | 63.14 | 10.08 | 30.04 | 97.68 | 45.95 | 57.96 | 42.64 | 95.99 | 62.00 | 75.30 | 83.26 | 72.59 | 69.20 | 28.80 | 39.54 | 71.35 | 29.18 |
| GDN | 16.47 | 26.84 | 17.87 | 65.30 | 9.62 | 39.08 | 84.15 | **53.25** | 69.09 | 42.18 | 38.04 | 71.12 | 49.53 | 76.89 | 24.10 | 68.95 | 45.61 | 52.64 | 82.39 | 39.43 |
| MAD-GAN | 15.31 | 21.39 | 17.63 | 63.31 | 10.53 | 30.80 | 92.42 | 46.08 | 63.07 | 43.75 | 84.83 | 69.53 | 76.42 | 86.63 | 63.69 | 79.97 | 49.00 | 60.76 | 81.07 | 49.02 |
| DiffAD | 10.51 | 19.89 | 13.75 | 60.34 | 7.67 | 27.76 | 100.00 | 43.45 | 55.09 | 33.04 | 12.02 | 100.00 | 21.46 | 18.71 | 7.27 | 46.76 | 16.53 | 24.37 | 80.96 | 20.74 |
| **Ours** | 17.06 | 41.26 | 24.10 | **79.82** | 15.10 | 41.55 | 58.60 | 48.60 | 65.42 | 43.28 | 96.20 | 66.92 | **78.92** | 86.91 | 74.81 | 65.42 | 62.63 | **63.99** | **93.40** | **49.22** |

Table 14: Standard deviations of anomaly detection point-based performance in Table 14. Standard deviations of threshold-dependent **P**recision, **R**ecall, F1 scores and threshold-independent AUC-ROC and AUC-PR scores are reported. All values are percentages.

| Method | SMD | | | | | PSM | | | | | SWaT | | | | | HAI | | | | |
|---|---|---|---|---|---|---|---|---|---|---|---|---|---|---|---|---|---|---|---|---|
| | P | R | F1 | $A_{ROC}$ | $A_{PR}$ | P | R | F1 | $A_{ROC}$ | $A_{PR}$ | P | R | F1 | $A_{ROC}$ | $A_{PR}$ | P | R | F1 | $A_{ROC}$ | $A_{PR}$ |
| IF | 1.16 | 2.02 | 0.47 | 0.72 | 1.50 | 3.06 | 9.03 | 1.54 | 1.12 | 1.70 | 0.22 | 0.11 | 0.03 | 0.91 | 0.86 | 1.43 | 2.09 | 0.59 | 0.67 | 0.90 |
| DIF | 4.06 | 0.98 | 0.98 | 0.50 | 1.05 | 1.59 | 1.06 | 1.65 | 1.52 | 2.38 | 1.87 | 0.73 | 0.60 | 0.34 | 0.56 | 1.91 | 3.22 | 2.62 | 1.15 | 1.94 |
| TranAD | 5.98 | 3.31 | 0.46 | 0.28 | 0.50 | 0.33 | 0.62 | 0.30 | 0.62 | 0.74 | 12.41 | 16.00 | 12.29 | 19.93 | 6.80 | 14.08 | 11.13 | 6.63 | 7.58 | 8.08 |
| ATF-UAD | 1.00 | 7.60 | 1.05 | 1.21 | 0.25 | 2.70 | 9.45 | 2.52 | 4.88 | 3.10 | 12.68 | 9.36 | 12.34 | 15.50 | 6.36 | 9.99 | 10.18 | 12.94 | 8.39 | 11.35 |
| AT | 0.00 | 0.00 | 0.00 | 0.44 | 0.43 | 0.00 | 0.00 | 0.00 | 1.95 | 1.01 | 0.00 | 0.00 | 0.00 | 5.39 | 1.35 | 3.33 | 3.13 | 3.18 | 5.81 | 1.59 |
| DCdetector | 0.00 | 0.00 | 0.00 | 0.78 | 0.13 | 0.00 | 0.00 | 0.00 | 0.32 | 0.15 | 0.00 | 0.00 | 0.00 | 0.04 | 0.21 | N/A | N/A | N/A | N/A | N/A |
| USAD | 2.10 | 1.91 | 1.17 | 2.68 | 1.20 | 0.44 | 0.43 | 0.50 | 0.61 | 0.27 | 3.30 | 1.41 | 0.70 | 2.64 | 1.25 | 13.69 | 6.17 | 4.12 | 4.73 | 5.44 |
| GDN | 7.60 | 12.09 | 3.46 | 0.58 | 2.19 | 2.50 | 11.27 | 4.26 | 2.87 | 2.06 | 1.75 | 1.49 | 1.19 | 1.50 | 1.56 | 14.18 | 11.91 | 5.65 | 1.60 | 4.23 |
| MAD-GAN | 2.46 | 2.16 | 1.08 | 1.98 | 0.52 | 2.31 | 5.94 | 2.08 | 4.54 | 5.54 | 0.91 | 0.93 | 0.60 | 2.44 | 2.22 | 0.86 | 0.90 | 0.87 | 1.42 | 1.27 |
| DiffAD | 0.38 | 0.61 | 0.28 | 0.34 | 0.39 | 0.00 | 0.01 | 0.00 | 0.74 | 0.45 | 0.00 | 0.00 | 0.00 | 0.20 | 0.03 | 5.14 | 0.87 | 1.24 | 0.77 | 0.47 |
| **Ours** | 0.76 | 4.22 | 1.20 | 0.56 | 0.51 | 1.65 | 0.65 | 1.06 | 0.79 | 1.09 | 1.44 | 1.27 | 0.93 | 0.48 | 1.11 | 0.38 | 0.38 | 0.23 | 0.60 | 0.62 |

# M  Point-based Evaluations

In addition to the model evaluations using range-based metrics such as VUS-ROC and VUS-PR in Sections 4.2 and 4.3, we conduct point-based evaluations herein using point-based metrics exclusively. Table 13 reports report the Precision, Recall, and F1 scores under the threshold where the method of interest achieves the best F1 score. To mitigate the impact of thresholding protocol, Table 13 also reports the threshold-independent AUC-ROC and AUC-PR scores. Table 14 reports the standard deviations.

SARAD achieves state-of-the-art performance on most datasets except PSM. The enlarged temporal receptive field of SARAD (half input window) contributes to its underperformance by point-based metrics, especially on datasets where anomalous ranges are short (see Table 2), when compared against most others' receptive filed of single or few time points (typical 1D Conv kernel size is 3). We again underline that most real-world anomalies are continuous and point-based metrics are mismatched for such anomalies (see Appendix I for discussion).

Table 15: Model complexity and overheads. The total training time (in minutes), the inference time per sample (in milliseconds), and the total number of parameters (where applicable) are reported. The best values are in **bold** and the second best underlined.

| Method | SMD $T=708K$ $N=38$ $\mathfrak{T}=1$ min Train. (mins) | Infer. (ms) | Param. | PSM $T=132K$ $N=25$ $\mathfrak{T}=1$ min Train. (mins) | Infer. (ms) | Param. | SWaT $T=497K$ $N=51$ $\mathfrak{T}=1$ s Train. (mins) | Infer. (ms) | Param. | HAI $T=922K$ $N=79$ $\mathfrak{T}=1$ s Train. (mins) | Infer. (ms) | Param. |
|---|---|---|---|---|---|---|---|---|---|---|---|---|
| IF | **0.01** | 0.01 | N/A | **0.00** | 0.01 | N/A | **0.01** | 0.01 | N/A | **0.03** | 0.01 | N/A |
| DIF | 10.13 | 1.66 | 874K | 1.91 | 1.52 | 853K | 7.67 | 1.68 | 895K | 15.47 | 1.87 | 940K |
| TranAD | 6.80 | 0.04 | 127K | 0.96 | 0.03 | 57K | 6.06 | 0.05 | 226K | 17.15 | 0.07 | 531K |
| ATF-UAD | 14.20 | 0.06 | 414K | 1.90 | 0.05 | 408K | 12.77 | 0.06 | 421K | 16.42 | 0.06 | 436K |
| AT | 0.74 | **0.00** | 867K | 9.25 | **0.00** | 4.80M | 36.53 | **0.00** | 910K | 60.57 | **0.00** | 4.91M |
| DCdetector | 102.16 | 0.01 | 867K | 25.72 | 0.03 | 895K | 214.56 | 0.02 | 910K | Out of Memory | | |
| USAD | 4.34 | 0.01 | 803K | 0.86 | 0.01 | 441K | 3.74 | 0.01 | 1.26M | 9.34 | 0.01 | 2.54M |
| GDN | 36.04 | 0.44 | **3K** | 9.64 | 0.34 | **3K** | 19.65 | 0.46 | **4K** | 65.69 | 0.68 | **6K** |
| MAD-GAN | 42.56 | 0.40 | 268K | 45.06 | 0.46 | 261K | 1416.26 | 0.40 | 274K | 165.01 | 0.47 | 289K |
| DiffAD | 11.93 | 1.49 | 38.85M | 4.94 | 7.91 | 38.85M | 15.32 | 1.69 | 38.85M | 42.87 | 2.80 | 38.85M |
| **Ours** | 1.47 | 0.11 | 9.57M | 2.32 | 0.12 | 15.85M | 10.87 | 0.16 | 9.59M | 31.97 | 0.39 | 15.94M |

## N  Model Complexity and Overheads

We study and compare the complexity and time overheads of all baselines and SARAD. Concretely, we evaluate the model complexity by the number of parameters being used and the total training time as well as the inference time per sample. All experiments on time overheads are performed on a compute node with AMD EPYC 7443 (48 cores, 96 threads) CPU, NVIDIA A10 (24GB) GPU, and 512 GB RAM.

Table 15 reports the total training time (in minutes), the inference time per sampling point, and the number of network parameters used (where applicable) of all baselines on the main datasets. The size of the training set $T$, the number of features $N$, and the sampling period $\mathfrak{T}$ are also reported per dataset for easy reference. SARAD, while not the fastest nor the lightest model, incurs moderate time overheads and model complexity.

We note that even SWaT, the smallest dataset in terms of actual clock time, spans approximately 6 days for collection. This is far exceeding most detectors' training time besides MAD-GAN and levigates concerns for training overheads for them. All detectors also incur an inference per sample time several magnitudes below the smallest sampling frequency of 1 second, guaranteeing real-time deployment of all detectors once trained.

# O    Baselines

We trained all baselines using official implementations where available and recommended hyper-parameters from respectively papers are used. Some baselines, such as DiffAD and AT, have dataset-specific hyperparameters and here we adopted them as well. The open-accessed URLs of the baselines used are listed as followed.

- IF (ICDM'08) (Liu et al., 2008): `https://github.com/xuhongzuo/deep-iforest`.
- DIF (TKDE'23) (Xu et al., 2023): `https://github.com/xuhongzuo/deep-iforest`.
- TranAD (VLDB'22) (Tuli et al., 2022): `https://github.com/imperial-qore/TranAD`.
- ATF-UAD (NN'23) (Fan et al., 2023): `https://github.com/wzhSteve/ATF-UAD`.
- AT (ICLR'22) (Xu et al., 2022): `https://github.com/thuml/Anomaly-Transformer`.
- DCdetector (KDD'23) (Yang et al., 2023): `https://github.com/DAMO-DI-ML/KDD2023-DCdetector`.
- USAD (KDD'20) (Audibert et al., 2020): `https://github.com/manigalati/usad`.
- GDN (AAAI'21) (Deng and Hooi, 2021): `https://github.com/d-ailin/GDN`.
- MAD-GAN (ICANN'19) (Li et al., 2019): `https://github.com/LiDan456/MAD-GANs`. Official implementation was migrated to pyTorch for uniform environmental set-up. See our codebase for details.
- DiffAD (KDD'23) (Xiao et al., 2023): `https://github.com/ChunjingXiao/DiffAD`.

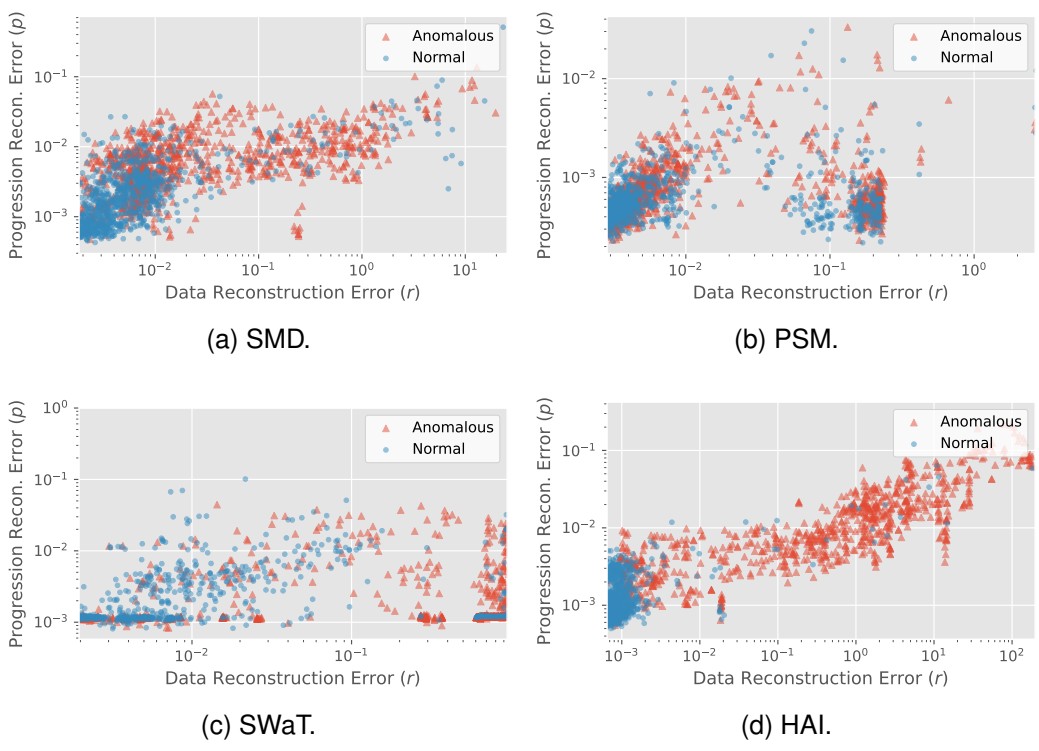

Figure 12: Joint detection criterion $s$ of data reconstruction error $r$ and progression reconstruction error $p$.

## P  Joint Detection Criterion

Figure 12 visualizes the two components of the joint detection criterion $s$ in Eq. 5, i.e., the data reconstruction error $r$ and the progression reconstruction error $p$. Balanced resampling is applied here. Recall that $s$ is the sum of normalized $r$ and $p$. Most anomalous samples (input series) either has high $r$ or high $p$, and oftentimes both. The former measures the magnitude of anomalousness in the data space, the latter in the spatial association space. The basis underpins the formalization of the joint detection criterion.

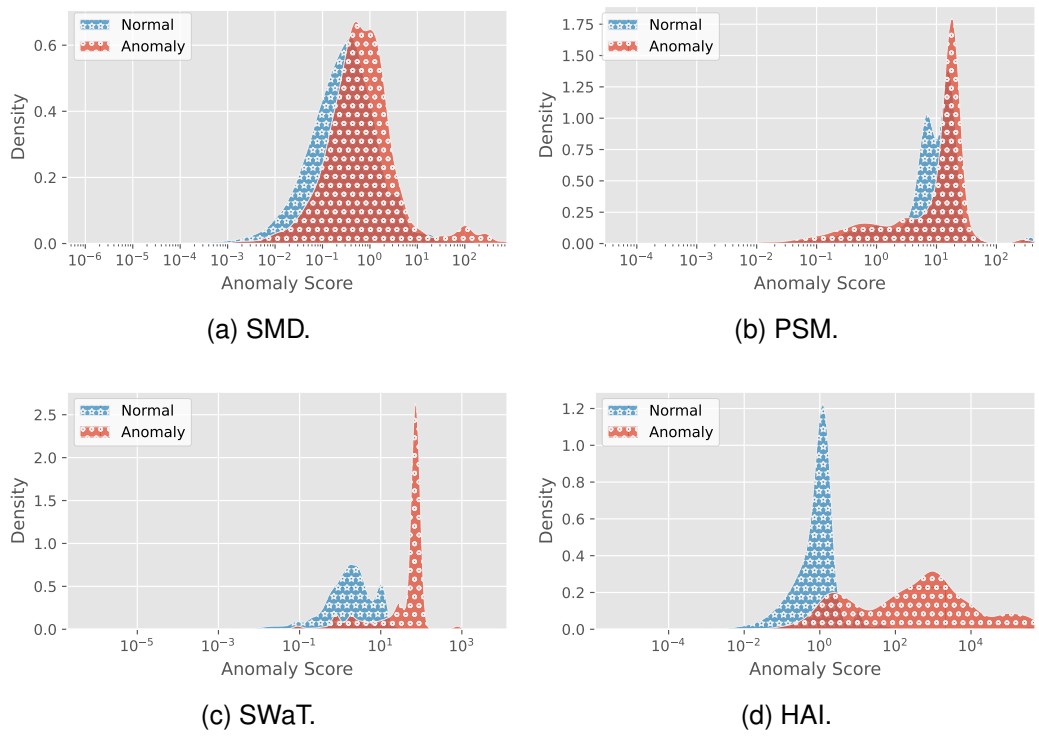

Figure 13: Distributions of anomalies scores.

## Q  Distributions of Anomaly Scores

Figure 13 the distributions of the joint anomaly scores. On SMD and PSM, the scores of the normal and anomalous samples (input series) overlap more heavily than on SWaT and HAI. The observations here highlight the difficulty of anomaly detection in the service monitoring space, more so than in industrial control where measurements and actuation result from well-defined control logics. The hardness is evident is Table 3 of detection performance and 4 of diagnosis performance.

# Table 16

Table 16: Anomaly detection performance under different thresholds. TPRs under thresholds set by pre-defined FPRs $\in \{1\%, 5\%, 10\%\}$ are reported. Higher TPRs are better. Similarly, FPRs under thresholds set by pre-defined TPRs $\in \{90\%, 95\%, 99\%\}$ are reported. Lower FPRs are better. All values are average percentages from five random seeds. The best values are in **bold** and the second best underlined.

| | SMD TPR↑(@FPR) | | | SMD FPR↓(@TPR) | | | PSM TPR↑(@FPR) | | | PSM FPR↓(@TPR) | | | SWaT TPR↑(@FPR) | | | SWaT FPR↓(@TPR) | | | HAI TPR↑(@FPR) | | | HAI FPR↓(@TPR) | | |
|---|---|---|---|---|---|---|---|---|---|---|---|---|---|---|---|---|---|---|---|---|---|---|---|---|
| Method | 1% | 5% | 10% | 90% | 95% | 99% | 1% | 5% | 10% | 90% | 95% | 99% | 1% | 5% | 10% | 90% | 95% | 99% | 1% | 5% | 10% | 90% | 95% | 99% |
| IF | 0.29 | 3.02 | 7.22 | 78.49 | 86.46 | 93.63 | 0.50 | 3.50 | 11.09 | 80.94 | 89.64 | 94.26 | 32.88 | 56.45 | 65.77 | 48.48 | 56.06 | 64.54 | 4.15 | 18.14 | 28.67 | 57.31 | 61.96 | 68.40 |
| DIF | 1.56 | 8.43 | 15.17 | 79.37 | 87.22 | 92.33 | 0.23 | 1.77 | 4.63 | 78.51 | 83.80 | 91.86 | 48.07 | **63.25** | **69.63** | **41.18** | 50.41 | 59.62 | 31.32 | 41.90 | 49.40 | 48.56 | 54.21 | 58.81 |
| TranAD | 0.86 | 4.25 | 8.64 | 87.86 | 91.53 | 95.21 | 1.15 | 2.58 | 7.62 | 75.94 | 87.46 | 96.24 | 0.12 | 0.57 | 1.61 | 75.78 | 81.11 | 85.72 | 21.39 | 31.48 | 37.79 | 53.60 | 58.50 | 66.99 |
| ATF-UAD | 0.04 | 1.04 | 4.42 | 90.65 | 93.75 | 96.61 | 0.43 | 2.05 | 5.67 | 82.90 | 89.02 | 95.95 | 0.23 | 1.02 | 2.16 | 69.66 | 75.93 | 81.20 | 16.93 | 24.21 | 31.12 | 60.42 | 65.08 | 69.25 |
| AT | 1.47 | 3.78 | 3.94 | 99.77 | 99.77 | 99.77 | 0.56 | 2.73 | 5.43 | 99.77 | 99.77 | 99.77 | 1.43 | 4.07 | 7.68 | 98.46 | 98.46 | 98.46 | 4.37 | 11.61 | 17.51 | 95.60 | 96.91 | 99.11 |
| DCdetector | 0.32 | 3.40 | 8.10 | 97.47 | 98.43 | 99.07 | 0.25 | 1.88 | 4.32 | 99.85 | 99.85 | 99.85 | 0.91 | 2.19 | 2.19 | 98.48 | 98.48 | 98.48 | N/A | N/A | N/A | N/A | N/A | N/A |
| USAD | 0.38 | 3.48 | 7.55 | 84.13 | 89.58 | 94.09 | 0.55 | 1.85 | 5.13 | 87.22 | 90.07 | 92.05 | 33.13 | 45.56 | 52.37 | 60.13 | 70.99 | 74.63 | 17.82 | 25.89 | 33.17 | 57.63 | 61.71 | 70.61 |
| GDN | 1.75 | 12.03 | 22.41 | 70.54 | 77.88 | 85.00 | 0.50 | 3.12 | 11.77 | **68.10** | **74.60** | **84.05** | 1.34 | 1.91 | 2.57 | 41.72 | 50.87 | 59.36 | 35.91 | 48.13 | 55.29 | 44.44 | 50.43 | 55.77 |
| MAD-GAN | **2.66** | 15.92 | 24.81 | 78.00 | 84.93 | 90.21 | **3.28** | **8.89** | **15.61** | 79.78 | 85.25 | 93.30 | 22.43 | 60.97 | 67.95 | 46.78 | 59.60 | 70.56 | 42.69 | 52.38 | 58.18 | 47.93 | 54.24 | 59.33 |
| DiffAD | 0.55 | 7.74 | 16.10 | 81.87 | 88.12 | 93.06 | 0.58 | 3.83 | 8.22 | 86.78 | 93.18 | 98.33 | 0.51 | 3.01 | 5.88 | 98.46 | 98.46 | 98.46 | 9.31 | 27.85 | 48.94 | 29.57 | 34.39 | 37.69 |
| **Ours** | 1.44 | **16.42** | **36.41** | **48.19** | **57.97** | **68.38** | 0.71 | 5.85 | 9.60 | 82.33 | 90.10 | 96.65 | **53.87** | 60.11 | 68.83 | 42.85 | 50.24 | 59.05 | **58.26** | **83.03** | **88.82** | **11.13** | **19.76** | **27.01** |

# Table 17

Table 17: Standard deviations of anomaly detection performance under different thresholds in Table 16. Standard deviations of TPRs under thresholds set by pre-defined FPRs $\in \{1\%, 5\%, 10\%\}$ are reported. Similarly, standard deviations of FPRs under thresholds set by pre-defined TPRs $\in \{90\%, 95\%, 99\%\}$ are reported. All values are percentages.

| | SMD TPR↑(@FPR) | | | SMD FPR↓(@TPR) | | | PSM TPR↑(@FPR) | | | PSM FPR↓(@TPR) | | | SWaT TPR↑(@FPR) | | | SWaT FPR↓(@TPR) | | | HAI TPR↑(@FPR) | | | HAI FPR↓(@TPR) | | |
|---|---|---|---|---|---|---|---|---|---|---|---|---|---|---|---|---|---|---|---|---|---|---|---|---|
| Method | 1% | 5% | 10% | 90% | 95% | 99% | 1% | 5% | 10% | 90% | 95% | 99% | 1% | 5% | 10% | 90% | 95% | 99% | 1% | 5% | 10% | 90% | 95% | 99% |
| IF | 0.08 | 0.52 | 0.46 | 1.34 | 1.82 | 0.91 | 0.09 | 0.84 | 1.00 | 3.07 | 1.03 | 2.23 | 4.29 | 4.61 | 2.39 | 2.68 | 2.88 | 2.74 | 0.51 | 1.04 | 1.11 | 2.00 | 1.46 | 2.98 |
| DIF | 0.48 | 0.90 | 1.44 | 1.27 | 1.45 | 1.48 | 0.16 | 0.66 | 0.38 | 2.11 | 1.73 | 1.91 | 4.64 | 2.05 | 2.18 | 1.48 | 0.78 | 0.84 | 2.91 | 3.61 | 3.00 | 2.37 | 2.74 | 2.17 |
| TranAD | 0.08 | 0.40 | 0.35 | 0.74 | 0.23 | 0.23 | 0.22 | 0.30 | 0.23 | 0.38 | 1.15 | 0.29 | 0.06 | 0.15 | 0.09 | 9.61 | 8.01 | 5.79 | 10.67 | 13.44 | 14.65 | 8.42 | 7.38 | 11.20 |
| ATF-UAD | 0.04 | 0.91 | 1.66 | 2.12 | 1.87 | 1.70 | 0.30 | 0.97 | 2.09 | 7.55 | 7.46 | 2.80 | 0.17 | 0.25 | 0.21 | 6.07 | 5.91 | 3.89 | 10.48 | 13.55 | 16.69 | 6.00 | 4.69 | 4.35 |
| AT | 0.06 | 0.51 | 0.67 | 0.00 | 0.00 | 0.00 | 0.05 | 0.53 | 0.91 | 0.00 | 0.00 | 0.00 | 0.62 | 0.81 | 1.31 | 0.00 | 0.00 | 0.00 | 2.24 | 4.63 | 5.58 | 3.32 | 2.06 | 0.00 |
| DCdetector | 0.04 | 0.61 | 1.39 | 5.15 | 3.00 | 1.57 | 0.03 | 0.42 | 1.04 | 0.00 | 0.00 | 0.00 | 0.02 | 0.60 | 0.60 | 0.00 | 0.00 | 0.00 | N/A | N/A | N/A | N/A | N/A | N/A |
| USAD | 0.22 | 0.74 | 1.26 | 3.34 | 3.14 | 3.31 | 0.12 | 0.00 | 0.01 | 0.28 | 0.44 | 0.34 | 9.68 | 11.59 | 8.33 | 15.06 | 14.38 | 15.27 | 7.29 | 10.77 | 11.55 | 6.59 | 5.71 | 5.00 |
| GDN | 1.51 | 5.42 | 4.85 | 1.77 | 2.13 | 2.75 | 0.42 | 2.05 | 2.87 | 8.93 | 8.37 | 5.87 | 1.55 | 1.57 | 1.72 | 3.10 | 6.71 | 7.48 | 12.76 | 7.33 | 5.96 | 3.83 | 4.28 | 3.06 |
| MAD-GAN | 1.23 | 1.54 | 1.30 | 4.22 | 4.57 | 3.55 | 1.74 | 2.58 | 3.12 | 6.60 | 6.30 | 2.91 | 19.27 | 5.15 | 6.39 | 15.81 | 11.37 | 11.48 | 1.92 | 2.97 | 2.88 | 4.87 | 4.66 | 4.20 |
| DiffAD | 0.11 | 0.75 | 0.52 | 1.18 | 0.74 | 0.23 | 0.09 | 0.28 | 0.26 | 0.98 | 0.48 | 0.29 | 0.07 | 0.32 | 0.34 | 0.01 | 0.00 | 0.00 | 0.48 | 2.26 | 0.99 | 1.98 | 2.66 | 2.54 |
| **Ours** | 0.18 | 1.76 | 4.81 | 0.48 | 0.87 | 1.35 | 0.35 | 0.80 | 0.97 | 1.42 | 1.25 | 0.73 | 3.57 | 3.14 | 0.45 | 0.70 | 1.08 | 1.69 | 1.51 | 2.45 | 2.73 | 2.80 | 2.36 | 2.12 |

# R  Detection Performance Under Thresholds

Effectiveness of each anomaly detector is influenced by the selection of thresholds. Previously, we reported the range-based and point-based threshold-independent performance metrics. Here, we study the influence of threshold selection on the detection performance. Table 16 reports the range-based True Positive Rates (TPRs) under thresholds set by pre-defined False Positive Rates (FPRs)as well as the FPRs under different thresholds set by pre-defined TPRs. Table 17 reports the standard deviations.

