# OpenReview forum: "SARAD: Spatial Association-Aware Anomaly Detection and Diagnosis for Multivariate Time Series"
_NeurIPS.cc/2024/Conference — NeurIPS 2024 poster_

### Official Review · Reviewer_KV18 · 2024-06-30

**Soundness:** 3
**Presentation:** 2
**Contribution:** 3
**Rating:** 5
**Confidence:** 3

**Summary:**

This manuscript proposes a spatial association reduction method for anomaly detection and diagnosis (SARAD) where anomalous features have low association values in reconstructing the multivariate time-series. The association values are derived from an attention between the features and exhibit changes with time which enables the detection of anomalous features and the time period of anomalies. SARAD is built on two models: a transformer for data reconstruction and an MLP for the reconstruction of spatial progression. The last anomaly score is a combination of both reconstruction scores.

**Strengths:**

- The proposed methods can be useful to detect the source of anomalies in multivariate time-series data.
- The manuscript provides a sufficient evaluation.

**Weaknesses:**

- In my opinion a main issue regarding the anomaly diagnosis should be addressed. For example, in Fig. 1 there are three features (9, 13 and 16) correlated with 12 and 15. An additional example is from Fig. 5 where features 13 and 16 are highly correlated with 12 and 15. What distinguishes the detected features from their correlated features?
- There was no discussion about the identity shortcut problem. Such a problem is typical for reconstruction-based methods where the model learns to reconstruct anomalies and thus it becomes harder to separate them from the normal ones. Maybe I missed it in the manuscript, but can you elaborate on this issue? How do you address this issue in your method?
- The Point-based evaluation on PSM dataset indicates some weaknesses regarding the sensitivity to the window size and short-term anomalies (see Table 13). The clustering for SMD and PSM is also not of the same quality compared to other datasets (see Figs. 12 and 13).
- The model size/parameters of the proposed method exceed other baselines by big margines (except for DiffAD).

**Questions:**

- How is the threshold defined from the anomaly scores to detect anomalies?
- Can you please provide a brief explanation why progression-based $p$ (SPR) is less effective for SWaT?

Minor:
- I find it a bit confusing to call it spatial association for time-series processing. Maybe it is better to call it channel/feature/variable.
- It is clearer if you visualize or indicate the ground truth for anomaly diagnosis alongside the detected features i.e., in Figs. 1, 3, and 4-7.
- Lines 84-85: Can you please make it clearer which spaces you are refereeing to? I think you mean data space and spatial progression space.

**Limitations:**

I think the issues about anomaly diagnosis and the identity shortcut should be discussed (see weaknesses).

---

> ### Author Rebuttal · Authors · 2024-08-05
>
> We would like to thank all reviewers for their detailed and constructive reviews. Here are our responses:
>
> **Weakness1:**
>
> **Response:**
> What distinguishes features 12 and 15 from features 9, 13, and 16 in Fig. 1 is that the spatial association reduction only occurs on the former features. Figs. 1(e)(f) shows that column-wise association reductions are significant (in darker cells) on features 12 and 15, but not on 9, 13, and 16. The reductions in Fig. 1e(f) are computed as the negative-only changes of the Transformer’s intermediate spatial association mappings from before (after) the anomaly to during the anomaly. Similarly, in Fig. 5, spatial association reduction occurs on features 12 and 15, but not on 13 and 16.
>
> **W2:**
> Identity shortcut happens when the data reconstructor learns to copy the anomalous input directly to the output. We considered two kinds of techniques to address the problem. On the data side, we could inject noises into the training input such that the reconstruction-based method does not know the true values. The added noise prevents the method from overfitting anomalies in the training set and learning to copy the input. On the model architecture side, we could apply self-attention masks within the MHSA module. In our case of SARAD, it prevents each feature’s representation from directly reading itself during attention computation.
>
> We initially implemented both but dropped them early in development because we did not run into the identity shortcut problem across all datasets. We also observed that, even in cases where the errors approach zeros (<1e-4), anomalies still lead to high anomaly scores relative to normal time points.
>
> **W3:**
> We acknowledged the weakness in Lines 688-693 on page 28. PSM contains many short-term anomalies (Fig. 8), making it hard to detect for all methods. In all our evaluations, we applied a unified time window length (of 100) to all datasets to offer a fair comparison with all baselines, as the majority of them originally or officially have done so as well.
>
> Fig. 10 and 11 show that with a smaller window size (of 10 or 20), SARAD achieves better performance than the default size. In deployment, the window length could be set based on the seasonal patterns, the sampling frequency, and (if available) previous anomaly history to maximize performance. We also note that nearly all methods are limited by the choice of window size.
>
> SMD and PSM are harder to detect anomalies upon compared to other datasets, leading to lower quality clustering in Figs. 12 and 13. The hardness is due to the existence of many short-term anomalies. PSM and SMD almost always have lower best performance scores than other datasets in Tables 3, 13, and 16.
>
> **W4:**
> In Lines 527-535, we acknowledged the limitations of the model size and complexity in regard to the number of features. The model size is modest for modern GPUs as we were training it on a three-year-old NVIDIA A10 GPU.
>
> Table 15 in Appendix N shows that, while our model size exceeds some other baselines, the time overheads are smaller than lightweight models such as GDN and MAD-GAN and comparable with many others as well, Anomaly Transformer and TranAD among them. In some cases, such as the DCdetector, the small model size does not translate into low time overheads due to its excessive contrastive learning pipeline.
>
> We also note that the inference times of our model are several magnitudes below all datasets’ sampling periods, guaranteeing real-time deployment. Also, the training times are several magnitudes below all datasets’ collection time, incurring little to no delay for model/project development.
>
> Finally, our backbone Transformer is known for fast parallel processing and we refactored MHSA implementation to enable parallel subseries processing (Line 168). Our design choice of the near vanilla Transformer and MLP architectures makes possible existing optimization techniques and a stripped-down loss function prunes excessive operations during loss computation.
>
> **Question1:**
>
> **Response:**
> While our main evaluation metrics are threshold-independent AUC scores, our method supports any threshold selection protocol. A common strategy would be to measure all anomaly scores $\boldsymbol{s}$ on the training/validation set and set the threshold to be $mean(\boldsymbol{s}) + 2 \cdot std(\boldsymbol{s})$.
>
> **Q2:**
> SWaT contains several extensively long-lasting anomalies, the longest being 35,900, during which most previously correlated features maintain their correlation due to the limited cyber-attack range (single attack point and single attack subprocess). While the SPR is indeed sensitive to changes in correlational relationships (similar observations are made in Fig. 3 example), the progression-based score decreases in the long anomaly span as the physical process regains stability. In addition, capturing SPR during a long-range anomaly is difficult as we are differentiating the association mappings between consecutive subseries within a fixed-length time window. The data-based score, however, captures the elevated or dropped steady-state measurements during an anomaly and thus complements its progression counterpart. In the end, we adopted a joint detection criterion that utilizes both.
>
> **Minor1:**
>
> **Response:**
> We note that spatiality has different connotations in some AI literature, e.g., geographic location or characteristics on Earth. We will mention such definitions in other related literature and with respect to that clarifications on our usage of the phrase “spatial association” in this work.
>
> **M2:**
> Ground truths for diagnosis labels are highlighted in red bounding boxes in Figs. 1, 3, 4, and 7. We will append the necessary descriptions.
>
> **M3:**
> Apologies for the unclear mentions of spaces. Yes, we are referring to both.

---

> > ### Comment · Reviewer_KV18 · 2024-08-10
> >
> > Thank you for your response. There are still two unanswered questions. I will try to rephrase my questions to make them clearer:
> >
> > **W1.** For example, in Fig. 5, the spatial association reduction happens for two features 12 and 15. The question is: Why didn’t this reduction happen for features 13 and 16 as well even though they are highly correlated with features 12 and 15?
> >
> > **W4.** My comment was about the model capacity rather than the training/inference time. As seen in Table 15, page 29, the proposed model exceeds other baselines by big margines (except for DiffAD). I think this gives the proposed model an advantage and doesn’t fully guarantee a fair comparison.

---

> ### Author Response · Authors · 2024-08-13
> **Response to W1 & W4**
>
> **W1:**
> You are correct in that there are non-anomalous features (9, 13, 16) correlating with anomalous features (12, 15) in Fig. 1. Similar correlated non-anomalous features can be found in Fig. 5. What distinguishes them from the anomalous features is the frequency of the patterns they showed during the anomaly period in the whole training set. Such patterns are frequent in the training set for features 9, 13, and 16, but not for features 12 and 15 (**see the following MSE experiment** for measuring frequencies). As the Transformer model was trained on the training set and learned the distributions of normal feature patterns, it does not register such frequent patterns of those correlated features as showing association reductions.
>
> **MSE Experiment:**
> We herein illustrate the frequencies of correlated non-anomalous features (9, 13, 16) in Fig. 1 relative to the anomalous (12, 15) using **Mean Squared Errors (MSEs)**. We first extract a subseries (of length 255) for each feature of interest during the anomaly period in Fig. 1 and refer to them as templates. We then compute the MSE between each template and each equal-length subseries from the training set (of size 566K) of the same feature. All features have already been normalized in data pre-processing. The table below shows the densities of MSE values for each feature on ranges starting from 0. It demonstrates a stark contrast between the anomalous features and the correlated non-anomalous features, with the majority of MSE values falling below 1 for the correlated and yet 0.00% for the anomalous features. The contrast remains stark as we shrink the ranges when plotting a histogram (which cannot be shown here due to OpenReview’s restrictions), with a dense concentration of MSEs near 0 for only the correlated features.
>
> |||||**MSE Range**||||
> |--------------|-----------|----------|----------|-------------|----------|----------|------------|
> ||**Feature**|**(0, 1]**|**(1, 2]**|**(2, 3]**|**(3, 4]**|**(4, 5]**|**(5, $+\infty$)**|
> |**Correlated**|9|**76.62%**|8.74%|4.60%|3.53%|2.46%|4.06%|
> ||13|**74.87%**|14.03%|3.58%|1.98%|1.01%|4.53%|
> ||16|**55.83%**|21.00%|7.60%|5.99%|2.71%|6.87%|
> |**Anomalous**|12|0.00%|1.43%|**90.45%**|3.66%|0.65%|3.81%|
> ||15|0.00%|**62.69%**|14.00%|11.88%|5.06%|6.36%|
>
> The MSE distributions observed here (and similarly for the Fig. 5 case) underline the usualness of correlated features’ patterns. Since other subseries in the training set share similar patterns for the correlated features, the Transformer model does not register those features as showing association reductions.
>
> **W2:**
> The large model size generally does not give our model an advantage in performance. We refer to our Hyperparameter Sensitivity experiments in Appendix H. Figures 10(c) and 11(c) both show that on most datasets (except SWaT) the performance is insensitive to exceedingly small attention length $D$ (also denoted as $d_{model}$ in some Transformer literature). The model size is largely decided by $D$ since the majority of parameters reside in linear transformations inside the Transformer's self-attention modules, whose weights are of shape $D \times D$. We chose 512 as the default value for $D$, following the original Transformer paper and subsequent works. Compared to the default value of 512, a small $D$ of 64 does not limit the model's performance on most datasets while scaling down the model by roughly 1/64, as shown in Figures 10(c) and 11(c). The following table reports the numbers of model parameters when $D=64$ and relative scales to the defaults when $D=512$ (as in Table 15). The model sizes are now comparable to most baselines, while the performances are of similar levels (except SWaT) as when $D=512$ and thus remain state-of-the-art.
>
>
> |                   | SMD    |       | PSM    |       | SWaT   |       | HAI    |       |
> | ----------------- | ------ | ----- | ------ | ----- | ------ | ----- | ------ | ----- |
> | Method            | Param. | Scale | Param. | Scale | Param. | Scale | Param. | Scale |
> | **Ours** ($D=64$) | 198K   | 1/48  | 284K   | 1/56* | 212K   | 1/45  | 343K   | 1/46  |
>
>
> One possible explanation for under-performance on SWaT is that SWaT has many long-range anomalies, which could require a large $D$ value to adequately encode rich spatial information during anomalies. Figures 10(c) and 11(c) show that $D=256$ (Param. count=2.46M, Scale=1/4) results in a much closer performance to the default $D=512$.
>
> ------
> *Correction: It came to our attention when computing the model scales that Table 15 in the original version incorrectly reported our parameter count on PSM. The count should be 15.85M, not 1.59M. Apologies for the typo. All other parameter counts for our model are correct. We will correct the count in the final version.

---

### Official Review · Reviewer_BjR8 · 2024-07-12

**Soundness:** 2
**Presentation:** 2
**Contribution:** 2
**Rating:** 5
**Confidence:** 4

**Summary:**

In this paper, the authors aim to address the problem of anomaly detection and diagnosis for time series data. The authors consider that the existing methods may obscure or dilute the spatial information and the interaction between different variates, they propose the SARAD model to capture these interactions with the transformer. The authors evaluate the proposed method on several datasets and achieve good performance.

**Strengths:**

N.A.

**Weaknesses:**

However, there are some problems to solve
1.	The authors claim that existing methods may assume feature independence or combine variables of a diverse physical nature. Indeed, the recent methods for time series analysis usually employ independent channel assumptions. However, several methods like the conventional RNN and the graph-based methods do not assume that the features are independent. Therefore, the motivation of this paper might not be convincing.
2.	Moreover, Figure 1 might be not clear to explain the motivation of the paper. Specifically, Figure 1(a) is not informative and Figure 1(b)(c)(d) are almost the same.
3.	The contribution of the proposed method is limited. There are other methods that use transformers as the backbone networks like [1]. It is suggested that the authors should provide a detailed discussion and compare with it.

[1] Xu, Jiehui, et al. "Anomaly transformer: Time series anomaly detection with association discrepancy." arXiv preprint arXiv:2110.02642 (2021).

**Questions:**

Please refer to weaknesses

**Limitations:**

Please refer to weaknesses

---

> ### Author Rebuttal · Authors · 2024-08-05
>
> We would like to thank all reviewers for their detailed and constructive reviews. Here are our responses:
>
> **Weakness1:**
>
> **Response:**
> We agree that some temporal modeling methods, such as RNN, do not make feature independence assumptions. However, Lines 28-34 stated that temporal methods including RNNs which do not assume feature independence combine variables/features of diverse physical nature. That leads to the dilution of spatial information, which could be crucial to detection. Our related work also covered RNN-based methods such as LSTM and MAD-GAN (Lines 91-94), the latter being one of our baselines. We highlighted their limitations regarding small receptive fields in time and the impact of timestamp misalignment across features (Lines 97-99).
>
> Our motivation for this paper is expressed in Lines 32-38: temporal methods ignore the spatial associations that characterize multivariate time series patterns. They also restrict diagnostic capabilities caused by a mismatch between temporal novelty and capturing spatial novelty. Our proposed SARAD aims to leverage spatial information and exploit the spatial association descending patterns common with time series anomalies.
>
> **W2:** Figure 1 investigates the Spatial Association Reduction (SAR) phenomenon by showing the changes in spatial associations throughout a time series anomaly.
>
> Fig. 1a shows the raw time series of all feature channels, with the anomalous features (#12 and #15) highlighted in red and marked with dashed lines. The intention of 1a is to show a full picture of the raw data before, during, and after the anomaly of interest. We were unable to insert proper space between features due to the page limit, which may cause readability issues without digital zoom. Based on the reviews, we can replace 1a with a selected feature set to improve readability.
>
> Figures 1(b)(c)(d) show the average association mappings before, during, and after the anomaly. While they might look similar, the differences between them are shown in Figs 1(e)(f) which display elevated association reductions on the anomalous features (#12 and #15). Apologies for the color scheme as it does not have enough contrast. To improve readability, we will increase the contrast for Figs. 1(b)(c)(d).
>
> **W3:** We discussed existing detectors of the Transformer backbone such as Anomaly Transformer and DCdetector in Lines 94-102 and 105-108. Furthermore, we compared against Transformer-based detectors including TranAD, Anomaly Transformer, DCdetector, and ATF-UAD in Tables 3, 4, 13, and 16.
>
> Concretely, for the cited Anomaly Transformer, we underlined its temporal modeling capabilities underpinned by the self-attention mechanism and yet restricted by the exceptionally small receptive field (typical 1D Conv kernel size is 3 for embedding) and cross-feature timestamp misalignment issues. For anomaly diagnosis, temporal methods such as Anomaly Transformer also mismatch the anomaly criterion of temporal novelty with spatial interpretation.
>
> We emphasize our contributions in this paper in Lines 80-87. Previous Transformer-based detectors fall short of spatial modeling. In both detection and diagnosis contexts, they ignore the spatial associations that ubiquitously characterize multivariate time series patterns. Specifically, compared with previous detectors using a Transformer backbone, our contributions are as follows.
>
> - Our method explicitly models spatial associations with self-attention modules to address the aforementioned receptive field and timestamp misalignment issues.
> - We design subseries division and refactor MHSA implementation accordingly to enable data shuffling during training and prevent memory storage of last association mappings.
> - We propose autoencoding in the spatial association space to exploit the association descending patterns common with anomalies and complement data autoencoding much less sensitive to spatial novelty.

---

> > ### Comment · Reviewer_BjR8 · 2024-08-12
> >
> > Thanks for the detailed response. The response has addressed my concerns. I have raised the score.

---

### Official Review · Reviewer_RKBC · 2024-07-15

**Soundness:** 3
**Presentation:** 2
**Contribution:** 3
**Rating:** 6
**Confidence:** 3

**Summary:**

In this paper, the authors focus on incorporating spatial information for time series anomaly detection and diagnosis. The proposed algorithm, SARAD, employs a transformer-based data reconstruction approach to capture inter-feature associations. By analyzing changes in these associations over time, the algorithm identifies anomalies under the assumption that anomalous features cause a reduction in the perceived association.

**Strengths:**

The idea of utilizing spatial association descending patterns for time series anomaly detection and diagnosis seems interesting.
The authors proposed new criteria for calculating anomaly scores.
Detailed experimental analysis has been performed to validate the proposed method.

**Weaknesses:**

The related work section lacks a detailed discussion of all the baseline methods used for comparison in result analysis. For a more comprehensive understanding and to highlight the contributions of the paperwork, it would be better to incorporate those methods and discuss their limitations.

The third paragraph in the Introduction section is not easy to follow.  Providing more background information and reorganizing would be recommended.

**Questions:**

I'm interested in understanding how the SARAD method distinguishes reductions in spatial associations that indicate anomalies from those that could be considered normal variations. Please clarify the types of anomalies the SAR phenomenon is effective for.

The authors could have enhanced their study by comparing SARAD with existing methods (e.g., InterFusion), which integrate both spatial and temporal information for anomaly detection

**Limitations:**

yes

---

> ### Author Rebuttal · Authors · 2024-08-05
>
> We would like to thank all reviewers for their detailed and constructive reviews. Here are our responses:
>
> **Weakness 1:**
>
> **Response:**
>
> Apologies for not including all baselines being compared in the related work section. Lines 89-102 introduced and discussed the limitations of several baselines including MAD-GAN (Li et al., 2019), ATF-UAD (Fan et al., 2023), USAD (Audibert et al., 2020), AT (Xu et al., 2022), and DCdetector (Yang et al., 2023). Lines 108-110 briefly mentioned GDN (Deng and Hooi, 2021) as a graph-based prediction approach for detection.
>
> We did not discuss IF and DIF baselines as they are based on neither temporal nor spatial modeling. That was our oversight. We will include a summary and limitations of IF and DIF in the final version. They work by building a binary decision tree ensemble that partitions either the data space (IF) or the deep embedding spaces (DIF). They are limited by the lack of temporal or spatial information and their anomaly scores are not reflective of the magnitude of the anomalies.
>
> We will also include TranAD, which replaces the MLP in USAD with Transformer, and its adversarial training paradigm makes the reconstruction errors more robust. However, it shares the temporal modeling nature with other Transformer-based detectors and is limited by the small receptive field to capture long-range inter-feature correlation and handle timestamp misalignment.
>
> With regards to GDN, we will also highlight its limitations beyond the lack of temporal changes of spatial associations mentioned, including the mismatch between its single-point prediction target and the range-wise anomalies and unstable Top-K node selection during training.
>
> **W2:**
> We will reorganize the third paragraph in the final version. Specifically, we will discuss the role of MHSA inside the Transformer and explain why the intermediate association mapping via MHSA captures the correlational relationships between features under our setting.
>
> **Question 1:**
>
> **Response:**
> SAR is effective whenever anomalies originate from or lead to the dissolution of pre-existing associations, detaching anomalous features from their non-anomalous counterparts (Lines 56-58). Such associations are common in industrial control systems and many other monitoring systems, in which sensors routinely collect measurements from different locations of interconnected subprocesses. Examples of SAR can be found in Figs. 1, 3, 4, 5, 6, and 7 where during the anomalies anomalous features experienced sudden unusual spikes, significant deviation from normal values, or some forms of correlational breakdowns with others. For example, in Fig. 7 a water pump P-101 (feature #4) was turned off without correlations with others causing an anomaly.
>
> To distinguish between anomalous SAR from normal variations, SARAD uses an MLP autoencoder on reduction in the association space (Lines 70-73). The autoencoder learns the patterns of normal variations by minimizing their reconstruction errors. It could dismiss normal SAR during inference, compared to using reductions directly (Lines 286-288 in Ablation Studies). In addition, SARAD’s data autoencoder produces higher scores for anomalies, complementing the shortcomings of the association autoencoder.
>
> Lines 113-119 stated that we derived our SAR insights from the cyber-physical defense literature, where time series anomaly detection is extensively applied (SWaT and HAI are both cyber-physical systems). There are dynamic watermarking approaches (Satchidanandan and Kumar, 2017 and Dai et al. 2023) that overlay actuation with randomized signals to exploit correlational breakdowns (SAR in some way) of attacks (anomalies) for detection. While they are intrusive defense approaches, our method is non-intrusive and yet still exploits SAR.
>
> **Q2:**
> We ran into some compatibility issues when we tried to set up InterFusion during the rebuttal period, i.e., InterFusion’s CUDA and TensorFlow versions are so old that our machines no longer support them. Unfortunately, we couldn’t sort out the problem by the rebuttal deadline. We will try our best to resolve the issues and provide InterFusion experimental results in the final version.

---

> > ### Comment · Reviewer_RKBC · 2024-08-12
> >
> > Thank you for the response. The authors have addressed all the issues menitoned in the review.

---

### Official Review · Reviewer_7FD6 · 2024-07-22

**Soundness:** 4
**Presentation:** 3
**Contribution:** 4
**Rating:** 7
**Confidence:** 5

**Summary:**

Paper proposes a deep learning based anomaly detection method for multi-variate time series data. The proposed method has two components in a single neural network. First component is a traditional auto encoder approach in which the input time series is broken into two parts (along time) and the model is trained to use the first part to generate the second part, and identify anomalies as a difference between the predicted and true output. The second component trains a transformer to generate the associations across the different features (akin to cross-correlation) and then train an MLP to predict the associations and identify anomalies if the predict associations are different than the observed associations. Thus, the method captures a broader range of temporal anomalies.

Experimental results on a variety of benchmark data sets and comparisons with state of art methods demonstrate that the proposed method (SARAD) is able to identify anomalies that might not be apparent to approaches that do not consider the associations.

**Strengths:**

Paper is well-described. The idea of using association seems novel, though I have seen papers using cross-correlation to understand the multi-variate effect.

Results are promising. The approach is compute intensive (scaling quadratic with the number of features), but authors do acknowledge the limitation and outline possible approaches to work around it.

**Weaknesses:**

A terminology issue - spatial often refers to data with positional coordinates (geographic or xy on a domain) which indicates spatial relationships between observations. On top of that, each observation would be represented as a multi-dimensional feature vector. In this paper, the authors use the term spatial to refer to the multi-dimensional feature vector representation of data. I found that very confusing. I think it might be better to fix the terminology to improve readability of the paper.

**Questions:**

Are there any types of anomalies that would still not be captured by this method?

Similar to above, what kind of false positives could be captured by adding the association part?

**Limitations:**

The paper has addressed the limitations - I have suggested a couple above. Paper does not have a direct societal impact so the authors have a reasonable response to that.

---

> ### Author Rebuttal · Authors · 2024-08-05
>
> We would like to thank all reviewers for their detailed and constructive reviews. Here are our responses:
>
> **Weakness 1:**
>
> **Response:**
> Apologies for causing the confusion. We note that spatiality has different connotations in some AI literature, e.g., geographic locations or characteristics on Earth. We will mention such definitions in other related literature and with respect to that clarifications on our usage of spatiality in this work to refer to the multi-dimensional feature vector of time series data. This terminology is also used in some related literature [1][2], i.e., time series prediction and anomaly detection.
>
> [1] Tryambak Gangopadhyay, Sin Yong Tan, Zhanhong Jiang, Rui Meng, and Soumik Sarkar. 2021. Spatiotemporal Attention for Multivariate Time Series Prediction and Interpretation. In ICASSP 2021 - 2021 IEEE International Conference on Acoustics, Speech and Signal Processing (ICASSP). 3560–3564.
>
> [2] Yu Zheng, Huan Yee Koh, Ming Jin, Lianhua Chi, Khoa T. Phan, Shirui Pan, Yi-Ping Phoebe Chen, and Wei Xiang. 2023. Correlation-Aware Spatial–Temporal Graph Learning for Multivariate Time-Series Anomaly Detection. IEEE Transactions on Neural Networks and Learning Systems (2023), 1–15.
>
> **Question 1:**
>
> **Response:**
> Very long-range anomalies are hard to capture even by our method. We note that on PSM and SWaT, there are a few very long-range anomalies lasting several thousand time points (the longest one is 35,900 as per Table 2). Over the long course of such anomalies, the anomalous features re-correlate with non-anomalous features, or the correlational breakdowns at the beginning of anomalies cease in the view of divided subseries, as the physical process regains stability. Table 6 shows the impact of those very long-range anomalies on SWaT performance when the score is entirely based on spatial associations (SPR). Without the cues from changes in spatial associations, our method relies on data reconstruction to detect such anomalies. In some cases, the system returned to normal operational states, leading to lower reconstruction errors and causing the anomalies to evade detection.
>
> **Q2:**
> False positives can occur when the correlational relationships between features are altered but not due to anomalies. Examples include situations in which the monitored system changes operational modes unexpectedly. On SWaT, we observe a few false positives which are suspected to be during changes of operational modes. However, due to a lack of operational scheduling details and setpoint (desired conditions) data during test set collection in the SWaT technical document, we were not able to verify that. On HAI, there is an operation task scheduler that periodically adjusts the setpoint values within the legal ranges to simulate benign scenarios. As these adjustments are periodical, there are fewer false positives on HAI due to mode changes. Table 16 shows the elevated false positive rates for SWaT when compared to HAI. Real-world possibilities of unplanned operational mode or setpoint changes are low.

---

> > ### Comment · Area_Chair_mvYT · 2024-08-13
> > **Thank you**
> >
> > We thank you for replying all questions raised by reviewer.

---

> > ### Comment · Reviewer_7FD6 · 2024-08-14
> >
> > Thanks for your responses. The clarifications are much appreciated. I stand by my current review rating.

---

### Decision · Program_Chairs · 2024-09-25

**Decision:**

Accept (poster)

**Comment:**

The authors proposed algorithm, SARAD, employs a transformer-based data reconstruction approach to capture inter-feature associations. By analyzing changes in these associations over time, the algorithm identifies anomalies. The strength of the paper lies in novel formulation, new anomaly score, and a Detailed experimental analysis. The weakness lies in lack of motivation, lack of clarity about contribution, and missing benchmarks. Most of them have been addressed in rebuttal. All reviewers are consensus about the acceptance of the paper after rebuttal.  Congratulations to the authors! I am requesting the author to carefully incorporate the suggestion provided by the review in the camera-ready version.